

# Carbon dioxide plume dispersion simulated at hectometer scale using DALES: model formulation and observational evaluation

Arseniy Karagodin-Doyennel [1,5], Fredrik Jansson [2], Bart van Stratum [3], Hugo Denier van der Gon [4], Jordi Vilà-Guerau de Arellano [3], and Sander Houweling [1,5]

[1]Department of Earth Sciences, Vrije Universiteit Amsterdam, 1081HV, Amsterdam, the Netherlands
[2]Faculty of Civil Engineering and Geosciences, Department of Geoscience and Remote Sensing, Delft University of Technology, Delft, the Netherlands
[3]Meteorology and Air Quality Group, Wageningen University Research, P.O. Box 47, 6700 AA Wageningen, the Netherlands
[4]Department of Air Quality and Emissions Research, TNO, 3584 CB Utrecht, the Netherlands
[5]SRON Netherlands Institute for Space Research, Utrecht, the Netherlands

**Correspondence:** Arseniy Karagodin-Doyennel (a.doyennel@vu.nl)

**Abstract.** Abstract. Developing effective global strategies for climate mitigation requires an independent assessment of greenhouse gas emission inventory at the urban scale. In the framework of the Dutch Ruisdael Observatory infrastructure project, we have enhanced the Dutch Large-Eddy Simulation (DALES) model to simulate carbon dioxide ($CO_2$) plume emission and three-dimensional dispersion within the turbulent boundary layer. The unique ability to explicitly resolve turbulent structures

at the hectometer resolution (100 m) makes DALES particularly suitable for detailed realistic simulations of both singular high-emitting point sources and urban emissions, aligning with the goals of Ruisdael Observatory. The model setup involves a high-resolution simulation (100 m × 100 m) covering the main urban area of the Netherlands (51.5°–52.5°N, 3.75°–6.45°E). The model integrates meteorological forcing from the HARMONIE-AROME weather forecasting model, background $CO_2$ levels from the CAMS reanalysis, as well as point source emissions and downscaled area emissions derived from 1 km × 1 km

emission inventory from the national registry. The latter are prepared using a sector-specific downscaling workflow, covering major emission categories. Biogenic $CO_2$ exchanges from grasslands and forests are interactively included in the hectometer calculations within the heterogeneous land-surface model of DALES. Our evaluation strategy is twofold, comparing DALES simulations with: (i) the state-of-the art LOTOS-EUROS model simulations and (ii) in-situ Cabauw tower measurements and Ruisdael surface observations of the urban background in the Rotterdam area at Westmaas and Slufter. Our comprehensive

statistical analysis confirmed the effectiveness of DALES in modeling the urban-scale $CO_2$ emission distribution and plume dispersion under turbulent conditions, but also revealed potential limitations and areas for further improvement. Thus, our new model framework provides valuable insights into emission transport and dispersion of $CO_2$, in support of emission uncertainty reduction using atmospheric measurements and the development of effective climate policies.

## 1   Introduction

Climate change is a critical global environmental problem caused by rising concentrations of carbon dioxide ($CO_2$) and other long-lived greenhouse gases (GHGs) (IPCC, 2021). To address this problem, international agreements like the Paris Agreement



aim to mobilize political forces to reduce GHG emissions. Expanding urban areas play a key role, as they account for 60-70% of global $CO_2$ emissions (IPCC, 2023). A recent United Nations Framework Convention on Climate Change (IPCC, 2023) report also highlights the prominent role of urban $CO_2$ emissions in amplifying climate change, underscoring the urgent
need to address them in mitigation efforts. However, urban environments pose challenges due to their complex, heterogeneous landscapes, diverse emission sources (e.g., transport, industry, biosphere interactions), and significant spatiotemporal variability caused by atmospheric turbulence. Tackling these challenges in the quantification of emissions requires high-resolution data to precisely identify emission hotspots, which is crucial for effective monitoring and mitigation.

To address the urgent question of how to reduce emissions most efficiently, many countries have developed national programs
for monitoring atmospheric GHG concentrations. Initiatives such as CarboCount-CH (see http://carbocount.wikidot.com/, last access: 27 November 2024) in Switzerland, the GAUGE project in the UK (Palmer et al., 2018), and the European ICOS initiative (see https://www.icos-cp.eu/, last access: 27 November 2024), the North American Carbon Program (see https://www.nacarbon.org/nacp/, last access: 27 November 2024) in the US, and the CONTRAIL project (https://cger.nies.go.jp/contrail/about/index.html, last access: 27 November 2024) in Japan, reinforce global efforts to establish transparent and accurate $CO_2$
and $CH_4$ emission tracking.

On the other hand, as cities are major $CO_2$ sources, targeted monitoring is becoming a priority. Due to their complexity and growth, cities require detailed observation and analysis, though monitoring them is particularly challenging (Huo et al., 2022). Programs like ICOS Cities (see https://www.icos-cp.eu/projects/icos-cities, last access: 27 November 2024), Urban-GEMMS (see https://www.arl.noaa.gov/research/atmospheric-transport-and-dispersion/urban-gemms/, last access: 27 November 2024), and the C40 Cities Climate Leadership Group support high-resolution modeling for capturing the fine-scale variability of urban emissions. Furthermore, the Megacities Carbon Project (see https://earthobservatory.nasa.gov/images/86970/megacities-carbon-project, last access: 27 November 2024) tracks emissions in global cities, supporting efforts to refine urban GHG inventories and strengthen mitigation policies (Timmermans et al., 2013).

In the Netherlands, there is a similar need. According to the Nationally Determined Contribution climate action plan, the
Dutch government aims to reduce $CO_2$ emissions by 55% by 2030 and achieve climate neutrality by 2050 (UNTC (United Nations Treaty Collection), 2016). Thus, comprehensive studies of urban emission sources and distribution in the environment are essential to meet these ambitious reduction targets. A notable initiative in this regard is the Dutch Ruisdael Observatory (see https://ruisdael-observatory.nl/, last access: 27 November 2024). This infrastructure project has been established to improve the accuracy of weather and air quality forecasts in a changing climate and provide society with this high-quality and highly
detailed information to address existing climate problems. One of the aims is to model the entire Dutch atmosphere at a 100m resolution, combining simulations with meteorological and atmospheric composition data.

Despite significant progress in emission modeling at different scales (Sarrat et al., 2007; Meesters et al., 2012; Liu et al., 2017; Super et al., 2017; Brunner et al., 2019; Jähn et al., 2020; Brunner et al., 2023), a critical lack of realistic modeling of urban-scale $CO_2$ emissions still remains. Moreover, capturing sub-kilometer emission plume features, such as dispersion and
inherent turbulence effects within the atmospheric boundary layer (ABL), might be important for accurate quantification of emissions. Hence, integrating anthropogenic emission inventories into frameworks like Large-Eddy Simulation (LES) models



(Deardorff, 1972), which explicitly resolve a major part of atmospheric turbulence, addresses this need. Brunner et al. (2023) demonstrated that LES models effectively capture $CO_2$ plume dynamics from coal-fired power plants, highlighting the importance of model resolution. Thus, despite the computational demands associated with LES, the development of such a simulation
framework has the potential to significantly enhance the capability of models to reproduce the observed $CO_2$ signal in urban areas (Sarrat et al., 2007; Liu et al., 2017; Super et al., 2017; Brunner et al., 2023). Along with that, incorporating dynamic biospheric models, which account for $CO_2$ plant assimilation and soil respiration, can further enhance urban-scale simulation by means of LES (Vilà-Guerau de Arellano et al., 2014).

To achieve high-resolution modeling, detailed emission inventories are essential. Previous studies have provided valuable in-
formation on various emission inventories at different scales, from global (Guevara et al., 2019, 2024) to regional (Urraca et al., 2024), across Europe (Xiao et al., 2021; Kuenen et al., 2022), Asia (Jia et al., 2021), and North America (Brioude et al., 2012), etc. In the Netherlands, for $CO_2$ emissions, the National Institute for Public Health and the Environment (RIVM) provides registered annual individual emission sources from the industry, as well as area emission inventory from various categories mapped on a km-scale grid (https://data.emissieregistratie.nl/, last access: 27 November 2024). Yet, they are not sufficient for
100m-scale LES models and cannot be employed without a proper downscaling. Yet, this process presents significant challenges due to spatiotemporal uncertainties that emerge when downscaling coarse-resolution data. For point sources, which are supposed to be easier to apply to LES due to their precisely available emitting locations, accurate vertical allocation through plume rise is crucial and not trivial to estimate, though it is important to account for in simulations (Brunner et al., 2019). Hence, achieving the required level of accuracy in emission modeling involves the complex processes of downscaling in space
and time, as well as accurate vertical allocation of emissions.

This need motivates the continued development of related improvements in LES tools and associated national emission inventories. One such model that is developed for the Netherlands is the Dutch Atmospheric Large Eddy Simulation (DALES) model framework (Heus et al., 2010; Ouwersloot et al., 2017). Traditionally, this simulation technique was employed primarily to study atmospheric physics and ABL dynamics (Heus et al., 2010; van Heerwaarden et al., 2017), but not to simulate $CO_2$
emission transport and distribution.

Thus, both having a high-resolution emission inventory and extending DALES with an advanced emission routine would enable us to realistically simulate the Dutch environment, aligning with the objectives of the Ruisdael Observatory research project.

This study addresses four main objectives:

1. **Document** the downscaling emission workflow program developed to prepare emission inventory for urban-scale realistic modeling of $CO_2$ emissions.

2. **Show** the capabilities of the state-of-the-art DALES 4.4 model, enhanced to simulate anthropogenic point sources and area-based $CO_2$ emissions, integrating biogenic $CO_2$ fluxes from vegetation.

3. **Validate** the framework and ability of DALES with the presented setup to simulate atmospheric $CO_2$ concentration
variability using observations and lower-resolution simulations, demonstrating the benefits of 100m-scale simulations.





4. **Assess** the importance of individual $CO_2$ components to unravel the overall $CO_2$ signal observed at measurement sites.

In reaching these goals, we provide valuable insights into the transport and dispersion of $CO_2$ plumes in turbulent environments. This enables us to quantify and evaluate emission inventories more accurately as well as investigate which scales should be resolved to adequately simulate the observed $CO_2$ concentration variability. This study is a step forward from the
initial work, introduced and discussed in (de Bruine et al., 2021).

The manuscript is structured as follows: Sect. 2 provides an overview of the anthropogenic emission datasets and the detailed description of the downscaled workflow to prepare emission model input. In Sect. 3, DALES and emission module code descriptions are provided. Sect. 4 details the model experiment setup. The datasets used for model validation are described in Sect. 5. Sect. 6 presents the model simulation results, their validation, and a discussion of the drivers behind the observed
variability. Finally, Sect. 7 and 8 provide an outlook on the further development of the tools and methodologies employed and summarize our study with general conclusions.

## 2 Anthropogenic Emission Data

Anthropogenic emission sources are classified into 10 groups according to the Standard Nomenclature for Air Pollution (SNAP). The SNAP categories used in this study are summarized in Table A1. We differentiate between two types of an-
thropogenic emissions: point sources and spatially allocated diffuse sources, which are processed in separate procedures as explained below.

Point sources, which include emissions from power plants and industrial facilities, are the largest contributors to the anthropogenic $CO_2$ budget, accounting for approximately 50–60% of total anthropogenic $CO_2$ emissions. In the Netherlands, companies responsible for these large emission sources are mandated to report emissions annually by location to a pollutant
register. Reported emissions from these sources are available at the national emission inventory (ER) portal, maintained by RIVM (https://data.emissieregistratie.nl/export, last access: 27 November 2024, hereafter referred to as ER portal). This portal provides an annual total emission inventory database of GHGs as well as other specific variables relevant for air quality can be acquired.

Emission data are classified by sector/subsector, facilitating processing for each SNAP category. Emissions from industrial
point sources are accessible at the ER portal, aggregated at the company level. A comprehensive list of registered emission sources, including thermal plume parameters like exhaust temperatures and volumetric flow rates, as well as stack height of emission itself, can be accessed from the RIVM upon request.

Besides, gridded $CO_2$ emissions with a spatial resolution of $1 \times 1$ km$^2$ over land and $5 \times 5$ km$^2$ over the North Sea are also available from the ER portal. The spatial resolution of $1 \times 1$ km$^2$ cannot be easily refined for all emission sources due to
various reasons; for some emissions there is a lack of suitable data to do so, sometimes privacy protection rules play a role. For most emission sources spatial allocation is done by applying an allocation key dataset, e.g. all emissions related to citizens are commonly gridded based on population number. Some total emissions are estimated with calculation methods in which spatial data is implemented/available (e.g. AIS data of ship movements), in this case only aggregation to the desired spatial scale is





needed. For industrial sources coordinates of stacks are registered in industrial activity surveys related to E-PRTR regulation.
An uncertainty of approximately 4% is reported for total $CO_2$ emissions in the emission inventory.

Comprehensive information on the methods used for the production and processing of the emission inventory, as well as uncertainties for different sectors, is provided in the National Inventory Report (Van der Net et al., 2024).

The emissions from the combined point sources and area sources utilized in our study represent the total national annual emissions, quantified in kilograms of $CO_2$ per year. Thus, these data need to be spatially and temporally disaggregated for use
in DALES. This is achieved through a downscaling workflow procedure, which will be described in the following section.

## 2.1 Emission Downscaling Workflow

Coupling the $CO_2$ emission inventory with a high-resolution model like DALES requires alignment between the spatiotemporal resolutions and coordinate systems of the emission inventory and the model. In this study, DALES input uses a 100 m spatial and an hourly temporal resolution. Therefore, to accurately simulate $CO_2$ emissions, the emission data must be disaggregated
in both space and time.

DALES is formulated on a rectilinear $x$-$y$ grid and configured to use the Arakawa C-grid (Arakawa et al., 2011, 2016). For this setup, Lambert Conformal Conic (LCC) coordinates are employed in DALES. Consequently, a translation of the coordinate system is required, as original emission datasets from the national registry are provided in Dutch Rijksdriehoek (RD) coordinates. Thus, we developed a downscaling workflow to process the prior emission inventory into DALES-compatible
input.

The workflow is structured as a comprehensive program with several stand-alone modules, each responsible for different aspects of emission data processing. Since DALES computes point sources and area emissions differently, the model input is separated into these two components. Initially, the workflow focuses on preparing point source data for DALES. For individual sources with precise emission locations, emissions are straightforwardly reassigned from RD to LCC coordinates.
Since DALES also calculates plume rise and emission altitudes interactively (discussed in Sect. 3.2), additional information on chimney height, exhaust temperature, and volumetric flux is required to calculate plume rise and the plume vertical borders, between which $CO_2$ is injected into the model atmosphere.

Unfortunately, not all point sources contain complete data. For instance, in the emission inventory for the year 2018, 1316 out of 1914 point sources had gaps in data, such as missing exhaust temperature, volumetric flow rate, or stack height. In this
case, a gap-filling approach is employed for those point sources using ordinary linear regression based on emission categories. This applies polynomial regression models to estimate missing or zero values in plume characteristics, based on the logarithm of emission values. For volumetric flow rate and stack height, linear regression models of polynomial order 1 are applied, with a logarithmic transformation for both. Temperature, which depends mainly on the emission process, uses a constant regression model (polynomial order 0) as it remains relatively stable across emission rates. Table 1 outlines these models and
their respective details.

Thus, the program returns the predicted values for the missing entries. Note that point sources with insufficient cohesive data available for regression are incorporated into area emissions.



**Table 1.** Gap-filling approach setup for different plume characteristics.

| Plume Characteristic | Regression Model | Polynomial Order | Transformation | Additional Details |
|---|---|---|---|---|
| Volumetric Flow Rate | Linear Regression | 1 | Logarithmic | Based on the logarithm of emission values |
| Exhaust Temperature | Constant Regression | 0 | None | Temperature is process-dependent; a constant value is used |
| Stack Height | Linear Regression | 1 | Logarithmic | Logarithmic transformation applied to stack height values |

Since the original emission data represent annual sums, temporal disaggregation down to the hour level is required. For this, we applied Emissions Database for Global Atmospheric Research (EDGAR) temporal profiles for anthropogenic emissions specified by SNAP category (TNO, 2011; Crippa et al., 2020). This accounts for emission variations at daily, weekly, and monthly time scales, capturing variations such as traffic rush hour patterns, seasonal differences in heating needs for households, and specific for different countries.

Unlike point source emission processing, the workflow procedure to prepare a high-resolution area emission inventory involves a more complex approach, as the exact coordinates of emissions are unknown. Initially, all point source emissions are subtracted from the area emissions, yielding "residual" area emissions. In essence, in the national emission inventory, area emissions include contributions from both diffuse sources (e.g., transportation, residential heating, agriculture) and point sources (e.g., industrial facilities, power plants). However, since we process point source and area emissions separately, it is essential to remove the point source contributions from the area emissions to avoid double-counting.

Subsequently, these residual area emissions are translated into GeoPackage format (gpkg). This format is chosen for its ability to precisely define spatial extent and select relevant subdomains within the Netherlands for simulation purposes. This eliminates the need for additional software (e.g., QGIS), reducing manual intervention.

The most computationally demanding operation in the workflow is the reprojection of area emissions to a high-resolution grid and from RD coordinates to the target LCC coordinates. The program intersects the RD grid cells with LCC coordinates and reassigns emissions to the new grid based on the proportional overlap of the grid box fractions. Emissions are then aggregated at the target resolution according to these proportions. This method is exact and scaling independent.

Figure 1 shows annually integrated $CO_2$ emissions for the year 2018, including area and point sources for each SNAP category.

In Figure 1, we show the contributions of different sectors to total $CO_2$ emissions. Over the sea, the resolution of input emissions is much coarser ($5\times5$ km$^2$) than over the land ($1\times1$ km$^2$), as evidenced by the large squares over the North Sea. Although point sources are barely visible at 100 m resolution (see Figure 2 for more detail), we present a combined view of area and point sources to provide a complete picture of emissions and verify the annual total within the selected domain. The



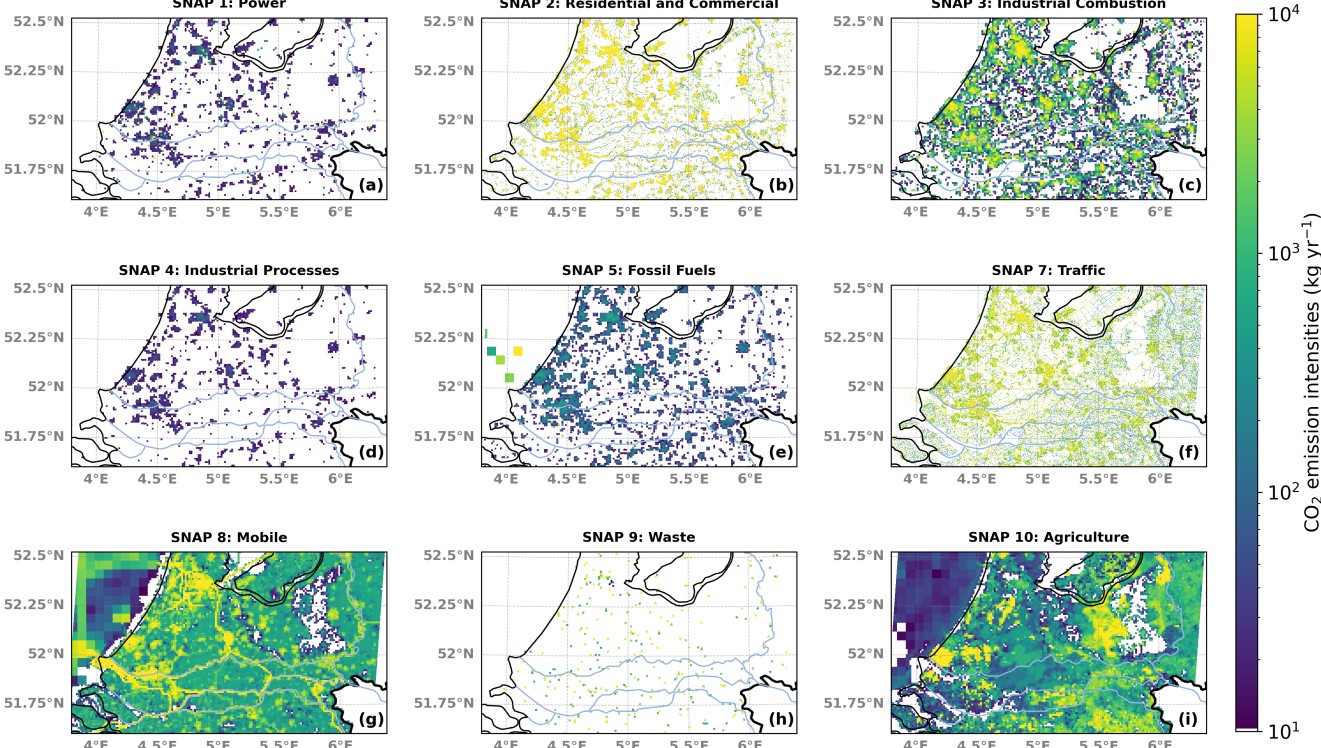

**Figure 1.** Annual surface $CO_2$ emission inventory (kg year$^{-1}$) over the simulation domain (51.5° - 52.5°N, 3.75° - 6.45°E; resolution 100 m) for the year 2018, categorised by SNAPs: **(a)** SNAP 1: Power; **(b)** SNAP 2: Residential and Commercial; **(c)** SNAP 3: Industrial Combustion; **(d)** SNAP 4: Industrial Processes; **(e)** SNAP 5: Fossil Fuels; **(f)** SNAP 7: Traffic; **(g)** SNAP 8: Mobile; **(h)** SNAP 9: Waste; **(i)** SNAP 10: Agriculture. These emission maps aggregate both area and point source emissions.

overall $CO_2$ emissions at hectometer resolution from all SNAP categories combined are presented in Figure 2. The total sum of these emissions is approximately 148 Mt/year for the selected domain, which aligns well with publicly available $CO_2$ emission estimates for the Netherlands in 2018 (Ruyssenaars et al. (2021); https://ourworldindata.org/co2/country/netherlands?country=

~NLD, last access: 27 November 2024). Since LES simulations are computationally expensive, the simulation domain only covers a part of the Netherlands. The selected domain includes the main focus area of the Ruisdael Observatory project (central part of the Netherlands), which is the most urbanized area of the country, responsible for the majority of carbon emissions. This ensures that all major $CO_2$ sources in the region are included in the domain. To minimize the influence of $CO_2$ surface fluxes from outside the domain, besides selecting a specific simulation domain, weather conditions with a stable northeasterly

wind were selected for model evaluation (see Sect. 5.1).

Further, the vertical distribution and plume rise height of area emissions are accounted for. In DALES, plume rise height for point source emissions is computed online using a special algorithm (Gordon et al., 2018; Akingunola et al., 2018) (described in Sect. 3.2). In cases of area emissions, the plume properties are unavailable, and the representation of area emission plume rise





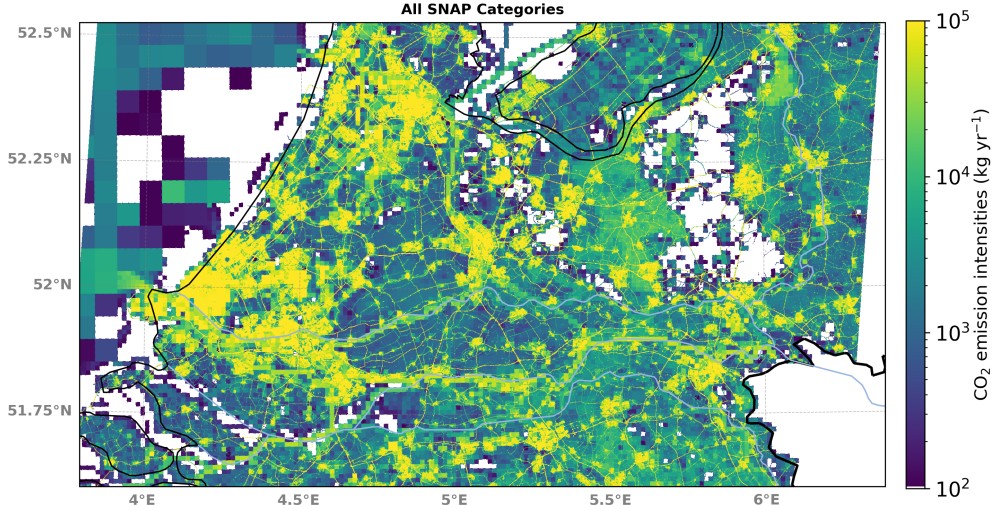

**Figure 2.** Annual surface $CO_2$ emissions (kg year$^{-1}$) at hectometer resolution aggregated across all SNAP categories within the simulation domain (51.5° - 52.5°N, 3.75° - 6.45°E; resolution 100 m) for the year 2018. These emission maps aggregate both area and point source emissions.

is simplified by setting the plume bottom to 0 and the plume top height to ∼150 m following Brunner et al. (2019). Emissions

are evenly distributed among model layers between the emission bottom and the top heights, so that each layer receives an equal share of the total emission values. It is important to note that area emissions from several SNAP categories have no vertical component, and all emissions from those categories are applied in the model at the ground level. These categories are: SNAP5, since fuel extraction occurs at ground level; SNAP7 traffic emissions; and SNAP10 agriculture, since it typically involves emissions at near-ground level, like those from soil and livestock.

Finally, annual emissions are disaggregated down to the hourly level using the EDGAR temporal profiles as discussed above for point sources. Note that the temporal integration time of DALES is approximately 2 seconds, so further inter-hour linear interpolation of emission input to smooth the hour-to-hour changes is necessary and applied directly in the model code (see Sect. 3.1). Final input files are date-specific and cover the complete simulation domain.

Figures 1 and 2 demonstrate the application of refinement methods using proxy or activity data for certain emission cate-

gories, which are explained further.

## 2.2 Refinement of area emissions: spatial disaggregation procedure

A spatial disaggregation procedure has been developed to refine $CO_2$ emissions for relevant categories using several high-detail activity data proxies, where such proxy data are applicable. In the current version of the workflow, proxy data for residential and traffic emission categories are applied. To refine residential emissions from the residential combustion (SNAP 2) category,

we employ demographic data from the Central Bureau of Statistics (CBS). Data have $100 \times 100$ m$^2$ resolution are freely





available from the CBS website

(https://www.cbs.nl/nl-nl/dossier/nederland-regionaal/geografische-data/kaart-van-100-meter-bij-100-meter-met-statistieken, last access: 27 November 2024). These datasets provide statistical information on a large number of parameters, including demographics, gas/electricity use, housing, energy, etc., for each $100 \times 100$ m$^2$ square across the Netherlands.

To refine area emissions from the SNAP 2 category, we use information about the average annual consumption of natural gas or total population density if gas usage is unknown. For refining the road transport (SNAP 7) category, we use a road shapefile containing detailed data on traffic intensity and nitrogen oxides ($NO_x$) emissions at the road level. This shapefile includes attributes such as the length of each road segment, and $NO_x$ emission intensities from light, medium, and heavy vehicles, respectively. These attributes provide essential information on emission intensity across different road segments. We utilise

the combined $NO_x$ emissions from these three vehicle types to determine the spatial distribution of traffic emission intensities within grids to derive $CO_2$ emission weights for road segments relative to traffic intensity. Thus, using these weights enables the refinement of $CO_2$ emissions from a $1 \times 1$ km$^2$ resolution to the target level. The traffic data shapefile is provided by the Dat.mobility company and can be requested from RIVM.

The refinement process for both SNAP 2: Residential and Commercial and SNAP 7: Traffic is illustrated in Figure 3.

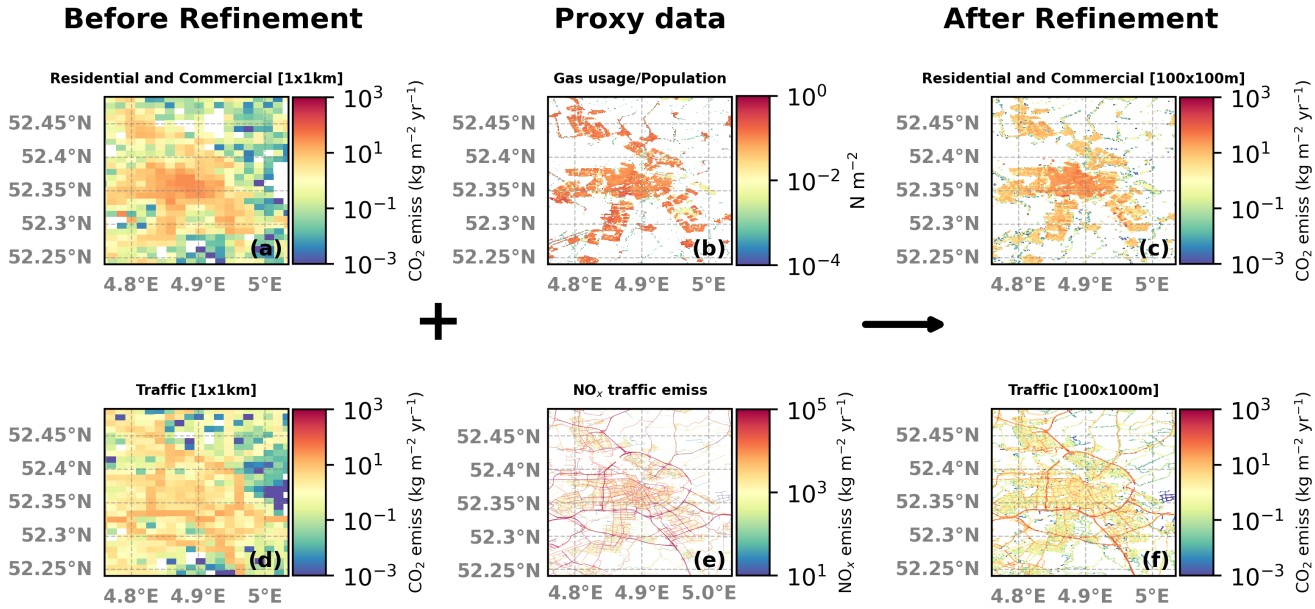

**Figure 3.** Illustration of the spatial redistribution of annual $CO_2$ area emissions (kg m$^{-2}$ yr$^{-1}$) from a coarse resolution of $1 \times 1$ km$^2$ to a finer resolution of $100 \times 100$ m$^2$ suitable for DALES, made for two SNAP categories: Residential Combustion (SNAP 2) and Road transport (SNAP 7). This illustration focuses on the area surrounding the city of Amsterdam. **(a)** and **(d)**: $CO_2$ emission fields (kg m$^{-2}$ yr$^{-1}$) at coarse resolution ($1 \times 1$ km$^2$); **(b)**: the gas usage/population density (N m$^{-2}$); **(e)**: Aggregate $NO_x$ emission data (kg m$^{-2}$ yr$^{-1}$) from three vehicle types: small, medium, and heavy; **(c)** and **(f)**: Resulting refined $CO_2$ emission fields (kg m$^{-2}$ yr$^{-1}$) at fine resolution ($100 \times 100$ m$^2$).





The importance of the refinement procedure lies in its ability to enhance the accuracy and specificity of emission locations. Figure 3 convincingly demonstrates that spatial information gained in the emission disaggregation process substantially improves the representation of emissions at the hectometer resolution required at the DALES numerical experiments. Without refinement, downscaling from 1 km to 100 m resolution would inaccurately retain 1 km shapes of objects, misallocating emissions. Utilising proxy data like $NO_x$ emissions and household statistic data further enhances the fidelity of emission downscaling, making estimates more spatially accurate in representing the real-world conditions.

Thus, the downscaling emission workflow program demonstrated a strong performance in preparing model emission inputs. Its easy-to-use interface ensures that the tool is available for a wide audience, and it is offered as a standalone resource for anyone interested. The workflow can easily be extended with other datasets and techniques for preparing high-resolution emission fields for any available atmospheric compounds in the future.

## 3   The DALES Model

The Dutch Atmospheric Large-Eddy Simulation (DALES) model is a community-based numerical framework designed for atmospheric research, particularly focusing on small-scale atmospheric turbulence processes, including clouds, and the physics of the ABL (Heus et al., 2010; Ouwersloot et al., 2017). DALES originates from the code developed by Nieuwstadt and Brost (1986). In this work, we use DALES version 4.4.

DALES is based on LES techniques, which resolve eddies in turbulent flow down to a certain scale (typically the size of the grid cells), below which small-scale turbulent structures are parameterized. Therefore, no parameterization of processes such as ABL entrainment or the mixing of plumes is required (Dosio et al., 2003). Additionally, DALES incorporates state-of-the-art atmospheric physics and microphysics schemes to simulate various processes, including radiation, convection, and cloud formation. These components are crucial for accurately representing the exchange of momentum, heat, moisture, and other substances between the atmosphere and the Earth's surface.

DALES is well-proven to accurately reproduce observed atmospheric turbulence and other dynamical processes, providing valuable insights into ABL phenomena, atmospheric dynamics, as well as cloud and aerosol microphysics (Sikma and Ouwersloot, 2015; de Bruine et al., 2019). Tracer advection is simulated using the mass-conserving Kappa scheme (Tatsumi et al., 1995). The Kappa scheme is a hybrid advection scheme that combines aspects of first-order upwind schemes and second-order centred schemes for parameters such as tracer mixing ratios that should never become negative. The filtered Navier-Stokes equations are solved on the DALES grid, allowing extremely fine spatial resolutions ranging from tens of metres to a few kilometres horizontally and from a few metres to several hundred metres vertically. DALES is developed for the troposphere; therefore, the vertical grid begins at ground level and extends to a height of about 11 km. DALES employs a temporal integration time as fine as 2 s.

DALES features an interactive land surface simulation, including photosynthesis and soil respiration, using the Land Surface Model (LSM) (Jacobs and de Bruin, 1997; Ronda et al., 2001; Balsamo et al., 2009; Vilà-Guerau de Arellano et al., 2014).



Involving the LSM is particularly valuable for studying the effects of land cover heterogeneity on atmospheric dynamics, microphysics, and ABL development, as well as atmospheric influences from the biosphere.

LSM provides DALES with the capability to compute net biogenic $CO_2$ fluxes, such as biospheric sinks through vegetation
photosynthesis and soil respiration fluxes. This is achieved in DALES using a dedicated scheme that integrates canopy and soil resistances based on the A-$g_s$ (net $CO_2$ assimilation rate (**A**), stomatal conductance (**$g_s$**) model. The performance of A-$g_s$ has been previously assessed, showing similar results to the widely-used Farquhar biochemical growth model (van Diepen et al., 2022). Initially proposed by Jacobs and de Bruin (1997) and subsequently refined and simplified by Ronda et al. (2001), the A-$g_s$ scheme adopted by DALES enables the calculation of stomatal conductances for both $CO_2$ and water, facilitating the
exchange of $CO_2$ between vegetation and the atmosphere. The transport of $CO_2$ into the leaf is the result of gross assimilation and dark respiration. Hence, the scheme incorporates a parameterization for soil respiration of $CO_2$ and the influence of soil moisture on canopy conductance.

Ultimately, the scheme provides insights into net $CO_2$ assimilation (photosynthesis) and soil respiration, accounting for factors such as temperature and vegetation type. While the A-$g_s$ scheme in DALES was primarily focused on grassland ecosys-
tems, this work enhances the scheme by incorporating parameters specific to forest. Parameters for the A-$g_s$ model used in DALES for both vegetation types are provided in Table A2. Separating forest from grasslands is especially important in the central-east region of the Netherlands, where forests are prevalent. This distinction improves the accuracy of computed $CO_2$ and momentum fluxes due to forest-specific surface roughness.

In our work, meteorological lateral boundary conditions (LBCs) in DALES are nudged toward data from the HARMONIE-
AROME mesoscale weather forecast model developed at KNMI (Bengtsson and Coauthors, 2017). To nudge DALES lateral boundaries to HARMONIE-AROME, we use a distinct dataset from the Winds of the North Sea in 2050 (WINS50) project (see https://www.wins50.nl/, last access: 27 November 2024, for additional details), which provides coverage over the Netherlands at an hourly temporal resolution (see https://dataplatform.knmi.nl/dataset/wins50-wfp-nl-ts-singlepoint-3, accessed: 27 November 2024, for further details). It uses common meteorological variables such as wind speed, wind direction, temperature,
pressure and relative humidity as well as sea surface temperature.

To incorporate background $CO_2$ concentration, LBCs are applied based on the Copernicus Atmospheric Monitoring System (CAMS) air quality forecast. These conditions are derived from CAMS delayed-mode analysis, which provides data for a geographical area spanning 50.5° to 54.0°N and 1.75° to 9.125°E at a resolution of 0.125° × 0.125° (~14 km), updating every 6 hours (for further details on CAMS output, visit https://atmosphere.copernicus.eu/, last access: 27 November 2024). LSM
also requires initial state data for initialization, encompassing parameters such as land use, soil inputs, vegetation properties, and van Genuchten parameters. This gridded input is derived from ERA5 data and subsequently translated to LCC coordinates.

Overall, it is expected that due to its accuracy, fine resolution, realistic modeling approach, and coupling with an emission inventory, DALES can conduct targeted simulations that isolate and examine specific aspects of atmospheric physics and dynamics under controlled conditions. Since DALES had not previously been utilized for modeling $CO_2$ mixing ratios, addi-
tional development was required to incorporate a program that converts and integrates emissions data for accurate horizontal representation as well as for vertical allocation in the model, as detailed in the subsequent section.



## 3.1 Emission module in DALES

To integrate and simulate the transport of anthropogenic emissions within DALES, we developed a module that converts emissions from kilograms of $CO_2$ per hour, calculated by the workflow, to $\mu g\ g^{-1}$, applying vertical allocation of emissions, inter-hour interpolation to integrate a smooth change of emission, and finally applying emissions to scalar $CO_2$ tracers. Currently, the scalar tracer for atmospheric transport of $CO_2$ in DALES is expressed in units of $\mu g\ g^{-1}$. The expression used to transfer area emission profiles into model scalar tracers is as follows:

$$CO_2 tracer_j = CO_2 tracer_j + \frac{area\_emis\_int_j}{3600 \cdot \rho_j \cdot dzf_j \cdot dx \cdot dy \cdot 1 \times 10^{-6}} \tag{1}$$

where $CO_2 tracer$ is the scalar $CO_2$ tracer [$\mu g\ g^{-1}$]; area_emis_int is the temporally interpolated 3-D field of emission input [kg hour$^{-1}$]; $\rho_j$ is air density [kg m$^{-3}$]; $dzf_j$ is the thickness of the full level [m]; dx and dy are grid spacing in $x$ and $y$ directions [m]; $1 \times 10^{-6}$ is the conversion factor from kilograms to micrograms; $j$ denotes the vertical layer index from 1 to $k_{emis}$, where emissions are allocated (for area emissions, $k_{emis}$ equals the closest layer to 150 m, according to the results of Brunner et al. (2019)).

The inter-hour interpolation factor (tfac) and temporal interpolation of emissions are calculated as follows:

$$tfac = \frac{mod(rtimee + 1800, 3600)}{3600} \tag{2a}$$

$$area\_emis\_int_j = (1.0 - tfac) \cdot emis\_past_j + tfac \cdot emis\_future_j \tag{2b}$$

where tfac represents the inter-hour interpolation factor [s]; rtimee is the elapsed time since the start of the simulation [s]; and emis_past and emis_future are the "past-modeltime" and "ahead-of-modeltime" 3-D fields of original emission input [kg hour$^{-1}$].

For selected individual sources, the plume bottom and emission altitude can be calculated interactively. The effective emission height can be significantly higher than the geometric height of a stack due to the buoyancy of the emission flux (Briggs, 1984). Therefore, plume rise is influenced by several factors, including stack geometry, flow properties (such as exhaust temperature and volumetric flow rate), and meteorological conditions (such as air temperature, wind speed, and atmospheric stability) (Brunner et al., 2019). To account for this in the simulation, the DALES emission module includes an online algorithm that calculates plume rise based on the interaction between model meteorology and source-specific data at each model time step. This algorithm will be described further.

## 3.2 Online Algorithm for Calculating Plume Rise Height

The algorithm implemented in DALES to calculate the plume height above the stack as well as the vertical boundaries of the plume after it has risen to equilibrium was originally proposed by Briggs (1984). We implemented a revised, up-to-date version of this algorithm, as outlined in Gordon et al. (2018) and Akingunola et al. (2018).





Initially, since the calculated stack height may not align exactly with a model grid point, the air temperature ($T_a$) and wind speed ($U_a$) at the stack height are determined from DALES data using linear interpolation. Once the atmospheric variables are obtained, the buoyancy flux ($F_b$) at the stack height, responsible for the updraft of turbulent eddies, is calculated based on the difference between the emission temperature ($T_s$) and $T_a$ using the following expression:

$$F_b = \begin{cases} \frac{g}{\pi} \cdot V_s \cdot \frac{T_s - T_a}{T_s} & \text{if } T_s > T_a \\ 0 & \text{if } T_s \leq T_a \end{cases} \tag{3}$$

where $F_b$ is the buoyancy flux; $g$ is the acceleration due to gravity (9.81 [m s$^{-2}$]); $V_s$ is the stack emission volumetric flow rate [m$^3$ s$^{-1}$]; $T_s$ is the emission temperature [K]; and $T_a$ is the air temperature at stack height [K]. This calculation indicates that the emitted plume is buoyant and rises only when $T_s$ exceeds $T_a$. The plume parameters are assumed to be in steady-state conditions as information about their temporal changes is unavailable.

Further, the residual buoyancy flux ($F_1$) is estimated based on atmospheric conditions and emission characteristics. With an iterative process, continuing until $F_1$ becomes non-positive, we compute the local stability parameter ($S_j$) for each subsequent model level as follows:

$$S_j = \frac{g}{T_{zhj}} \left( \text{grad}T + \frac{g}{c_p} \right), \tag{4}$$

where $j$ represents the index for the half-grid level above the emission stack height; $c_p$ is the specific heat capacity of air, equal to 1005 [J K$^{-1}$ kg$^{-1}$]; and gradT is the air temperature gradient, calculated from air temperature and altitude differences as

$$\text{grad}T = \frac{T_{zhj+1} - T_{zhj}}{z_{j+1} - z_j}. \tag{5}$$

Note that at the initial step of the iteration, $T_{zhj}$ and $z_j$ correspond to $T_a$ and the stack height ($h_s$), respectively.

The $F_{1,j+1}$ is calculated sequentially for each atmospheric layer based on the value of $S$, selecting the final value that shows the greatest decrease in flux, as recommended by Briggs (1984), as follows:

$$F_{1,j+1} = \begin{cases} \min \left( F_{1,j} - 0.015 \cdot S_j \cdot F_{1,j-1}^{1/3} \cdot ((z_{j+1} - h_s)^{8/3} - (z_j - h_s)^{8/3}), \\ F_{1,j} - 0.053 \cdot S_j \cdot U_m \cdot ((z_{j+1} - h_s)^3 - (z_j - h_s)^3) \right), & \text{if } S_j >= 0 \\ F_{1,j}, & \text{if } S_j < 0 \end{cases} \tag{6}$$

where the mean wind speed $U_m$ is calculated as $(u_{zhj+1} + u_{zhj})/2$, as recommended by Gordon et al. (2018). In the first iteration, $u_{zhj} = U_a$ and $z_j - h_s = 0$. The stack height is subtracted from each $z$ value, representing the vertical distance relative to the top of the stack. Initial values for $F_1$ are set as $F_{1,j-1} = F_{1,j} = F_b$.



Finally, the exact plume rise height ($h_{\mathrm{max}}$) is determined based on the condition that $F_{1,j+1}$ at $h_{\mathrm{max}}$ equals 0, indicating that $h_{\mathrm{max}}$ is the altitude at which the buoyancy flux of emitted plume dissipates entirely (Akingunola et al., 2018). Thus, the expression for $h_{\mathrm{max}}$ can be derived from (6) and $F_{1,j+1} = 0$ and applied in the layer where $F_{1,j+1}$ becomes negative, as follows:

$$h_{\mathrm{max}} = \begin{cases} \min\left(\dfrac{F_{1,j}}{(0.015 \cdot S \cdot F_{1,j-1}^{1/3})^{3/8}} + (z_j - h_{\mathrm{s}}), \right. & \\ \left. \dfrac{F_{1,j}}{(0.053 \cdot S \cdot U_{\mathrm{low}})^{1/3}} + (z_j - h_{\mathrm{s}})\right), & \text{if } F_{1,j+1} < 0 \\ z_{j+1} - h_{\mathrm{s}}, & \text{if } F_{1,j+1} = 0 \end{cases} \tag{7}$$

If $F_{1,j+1} = 0$, then $h_{\mathrm{max}}$ equals the altitude of $F_{1,j+1}$.

Using the plume rise height $h_{\mathrm{max}}$, the top ($z_{\mathrm{t}}$) and bottom ($z_{\mathrm{b}}$) of the plume are then calculated as follows:

$$z_{\mathrm{b}} = h_{\mathrm{s}} - h_{\mathrm{max}} \cdot 0.5 \tag{8a}$$
$$z_{\mathrm{t}} = h_{\mathrm{s}} + h_{\mathrm{max}} \cdot 1.5 \tag{8b}$$

The illustration of resulting plume top distributions for midday (0:00 UTC) and midnight (12:00 UTC) times are depicted in Fig. 4.

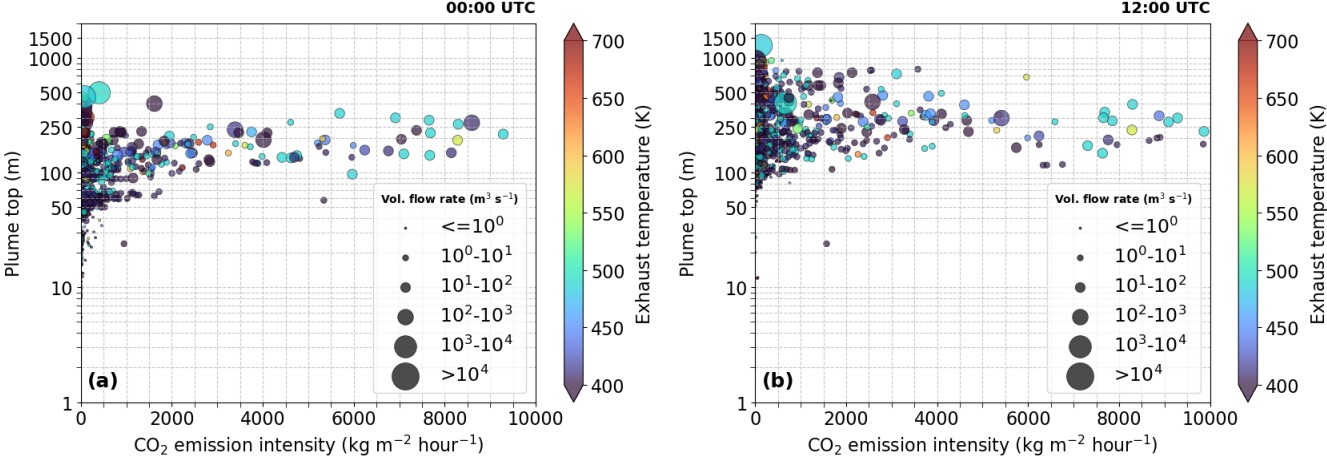

**Figure 4.** Modelled plume top ($z_{\mathrm{t}}$) distribution as a function of the corresponding $CO_2$ emission intensity (kg m$^{-2}$ hour$^{-1}$) at **(a)** 0:00 UTC and **(b)** 12:00 UTC. Dot colour: exhaust temperature (K); Dot size: volumetric flow rate (m$^3$ s$^{-1}$).

Despite the exhaust temperature and volumetric flow rate remaining constant in the algorithm, a pronounced difference in plume top distributions between night (00:00 UTC) and day (12:00 UTC) times is visible. This difference is primarily due to local atmospheric conditions. During the night (00:00 UTC), the plume tops are confined below 500 m. In contrast, during the day (12:00 UTC), the plume tops exhibit greater variability, with some tops reaching up to 1500 m.





At night, the atmosphere is more stably stratified, with little turbulence and reduced vertical mixing. This stable stratification acts as a natural barrier, preventing plumes from rising higher into the atmosphere. Additionally, the boundary layer is lower at night, further constraining the height of plume rise. In contrast, during the daytime, solar heating causes surface warming, leading to increased atmospheric turbulence and stronger vertical mixing. This creates a deeper and more unstable boundary layer, spurring plumes to rise higher. The convective upflow during the day enhances the buoyancy of plumes, contributing to the broader distribution of plume tops observed at 12:00 UTC. Hence, the difference in plume top heights between night and day is largely driven by variations in atmospheric stability, turbulence, and boundary layer dynamics.

It is important to note that, as with area emissions, point source emissions are equally distributed vertically from plume bottom to plume top with grid cells fully covered by the plume. However, since the parameterization provides the exact plume bottom and plume rise heights, these altitudes may fall between the edges of model layers. The fractions of layers covered by the plume for the plume top ($z_{\mathrm{t,frac}}$) and bottom ($z_{\mathrm{b,frac}}$) are calculated as follows:

$$z_{\mathrm{t,frac}} = \frac{z_{\mathrm{t}} - z_{\mathrm{h}_{i_{zt}}}}{\mathrm{dzf}_{i_{zt}}}, \tag{9a}$$

$$z_{\mathrm{b,frac}} = \frac{z_{\mathrm{h}_{i_{zb}+1}} - z_{\mathrm{b}}}{\mathrm{dzf}_{i_{zt}}}, \tag{9b}$$

where $i_{zt}$ and $i_{zb}$ are the indices of layers that are higher or equal to $z_{\mathrm{t}}$ and $z_{\mathrm{b}}$, respectively.

To include the point source emission profile into the scalar $CO_2$ tracer and account for the vertical allocation of emissions using the calculated plume vertical boundaries, we use a similar expression as in Eq. (1), but applying it separately for three cases: $z_{\mathrm{b}} = z_{\mathrm{t}}$, $z_{\mathrm{b}} - z_{\mathrm{t}} = 1$, and $z_{\mathrm{b}} - z_{\mathrm{t}} > 1$, respectively. The full expression of point source emissions integrated into the model tracer can be found in (B1).

Thus, the use of the plume rise algorithm ensures a more accurate representation of $CO_2$ plume vertical distribution, contributing to a more realistic dispersion of pollutants. The plume rise height is strongly influenced by turbulent conditions and variations in buoyancy flux (see Fig. 4), which are calculated based on the differences between plume thermal parameters, ambient meteorology, and atmospheric stratification and stability.

## 4 Experiment design to budget the $CO_2$ contributions

### 4.1 Systematic experiments $CO_2$

To assess the contributions of different sources to the overall $CO_2$ concentration based on their origin, we devised a comprehensive model experiment featuring four distinct passive scalar $CO_2$ tracers with the following setups:

– **$CO_2$bg**: Represents the background concentration derived from CAMS.

– **$CO_2$bg_emiss**: Combines the background concentration with all anthropogenic emissions.

– **$CO_2$bg_emiss_resp**: Combines the background concentration, anthropogenic emissions, and net soil respiration.



– **CO$_2$sum**: Combines all contributions of atmospheric CO$_2$ included in DALES: CO$_2$bg, CO$_2$emiss, CO$_2$resp, and the net CO$_2$ assimilation (**CO$_2$photo**).

The **CO$_2$bg** tracer uses CO$_2$ molar fractions from CAMS, reprojected onto the DALES domain boundaries. **CO$_2$sum** in DALES is the final CO$_2$ tracer, which can be compared to observations, as it includes all considered components of atmospheric CO$_2$ variability. Note that LSM uses the **CO$_2$sum** tracer to calculate the ambient CO$_2$ mixing ratio.

The impact of photosynthesis can be isolated by subtracting **CO$_2$bg_emiss_resp** from **CO$_2$sum** due to the linearity of passive tracer transport in DALES. This experimental setup allows all individual components of CO$_2$ variability to be derived and evaluated separately.

As mentioned above, in this experiment DALES is configured with the simulation domain spanning from $51.5°$ to $52.6°$N and from $3.75°$ to $6.45°$E, with a horizontal resolution of $100\,\mathrm{m}$ (see Sect. 2.1).

### 4.2 Selected period of simulation

We assess the capability of DALES to simulate CO$_2$ variability for the summer period from 25 June 2018 to 28 June 2018. The selection was done based on the availability of model input data (particularly for nudging the large-scale meteorology) and the CO$_2$ measurements for validation.

The period was characterized by stable summer conditions, with predominantly clear skies and relatively warm temperatures. Winds were light to moderate, with a prevailing northeasterly flow (see Fig. 5), contributing to weak atmospheric mixing during the nighttime and early morning hours. These meteorological conditions were ideal for evaluating CO$_2$ variability, facilitated the detection of both anthropogenic emissions and biogenic fluxes.

To ensure model stability, the first day of the simulation period was excluded from the analysis to provide sufficient initialization time for the modelled physics and dynamics, as suggested by Savazzi et al. (2024). Although excluding the whole first day from validation may be considered excessive for the domain size covered by DALES, it ensures that biases related to model initialization are significantly mitigated.

## 5 Model evaluation data

### 5.1 In-situ observations

To evaluate and validate the developed modeling framework, we use data from several measurement sites across the Netherlands (see Fig. 5). The longest measurement time series is from the Cabauw tower ($51.9703°$N, $4.9264°$E). The Cabauw Atmospheric Observatory measures atmospheric CO$_2$ concentrations at four distinct elevations: at 27, 67, 127, and 207 m above the ground. These measurements are essential for a comprehensive characterization of local vertical gradients of CO$_2$ within the lower ABL, enabling detailed investigations into the vertical distribution and temporal variability of GHGs. The hourly-averaged CO$_2$ mole fraction data from Cabauw (Hazan et al., 2016) are freely available from the ICOS Carbon Por-



tal (https://data.icos-cp.eu/portal/ last access: 27 November, 2024) for both ICOS and non-/pre-ICOS periods (Frumau et al., 2024a, b, c, d).

Besides, we use hourly-averaged in-situ measurements of near-surface atmospheric $CO_2$ concentrations around the city of Rotterdam at two urban background stations, which are part of the Dutch Ruisdael Observatory, located at Westmaas (51.786°N, 4.45°E, Sampling height is 10 m) and Slufter (51.933°N, 3.999°E, Sampling height is 10 m) near the shore of North Sea (see Fig. 5).

These locations allow us to assess the model performance for the different contributions to $CO_2$. Slufter is located on the
2e Maasvlakte, close to the shore of the North Sea, while Westmaas resides south of Rotterdam in an agricultural area. The dominant wind direction in the Netherlands is from the South-West, so on many occasions both stations are upwind from the industrial-urban complex of the Rotterdam Rijnmond area. Yet, during the simulated period, the average wind direction was from the North-East (see Fig. 5), when Westmaas is located downwind of Rotterdam. The Cabauw tower is situated in a rural area with a mixed contribution of vegetation (generally grassland) and dispersed anthropogenic emissions, originating either
from dispersed local area sources or distant urban areas.

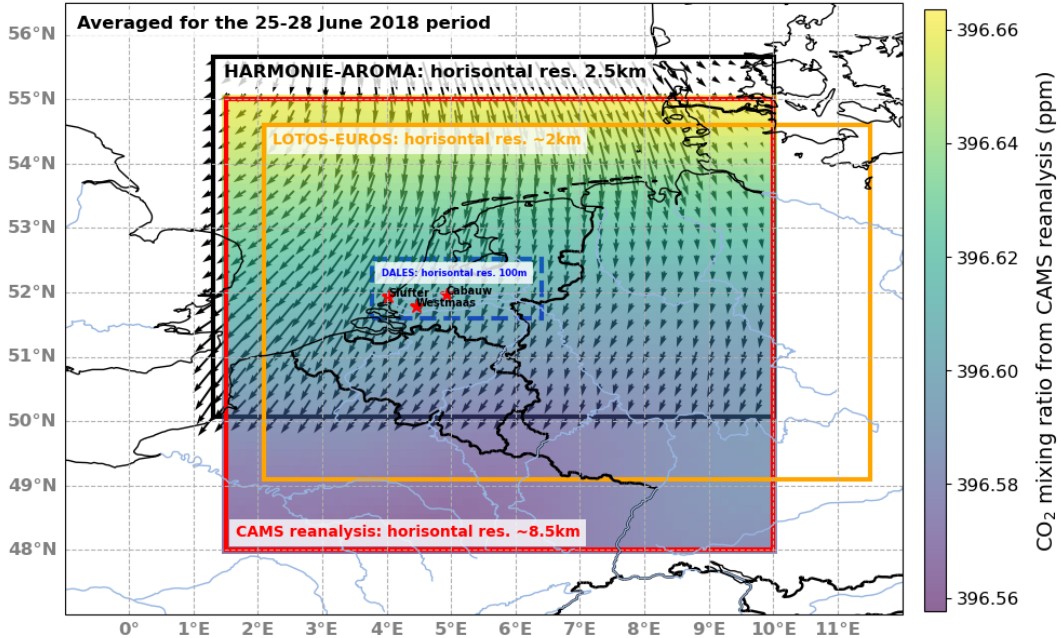

**Figure 5.** A map of the Netherlands with the measurement sites used to assess the simulations of atmospheric $CO_2$. The map shows surface $CO_2$ mole fractions from CAMS, averaged over June 25-28, 2018. Red stars indicate measurement locations, from left to right: Slufter (51.933°N, 3.999°E); Westmaas (51.786°N, 4.45°E); and the Cabauw tower (51.9703°N, 4.9264°E). The blue dashed line represents the borders of the DALES domain. The orange rectangle represents the borders of the LOTOS-EUROS domain. The black rectangle represents the borders of the HARMONIE-AROMA domain. The arrows indicate the wind direction averaged over June 25-28, 2018. The length of the arrows represents the wind speed.



## 5.2 LOTOS-EUROS simulation

The LOTOS-EUROS chemistry-transport model is used in comparisons with DALES to assess the added value of using a high-resolution turbulence resolving model for simulating the observed variability in atmospheric $CO_2$ in the Randstad area. LOTOS-EUROS is a state-of-the-art regional scale community model developed jointly by TNO and RIVM (Schaap et al., 2008; Manders et al., 2017), designed to simulate the dispersion and transformation of pollutants in the atmosphere, including aerosols, ozone, and trace gases. It accounts for anthropogenic and biogenic emissions, using complex chemical and physical processes for detailed forecasts of air quality and trace gas concentrations. The model links emission sources to atmospheric processes and transport, offering a comprehensive view of pollutant behaviour and distribution (for more information about LOTOS-EUROS, visit https://airqualitymodeling.tno.nl/lotos-euros/, last access: 27 November, 2024).

For our study, LOTOS-EUROS was employed to model $CO_2$ variability with a finer resolution than its standard configuration. Although the standard resolution of LOTOS-EUROS is approximately $25 \times 25$ km$^2$, the simulations used in this work were performed at a horizontal resolution of $\sim$2 km. Turbulence is parameterized in LOTOS-EUROS, enabling us to evaluate the benefits of using explicit turbulence in comparison with DALES. While the vertical resolution of LOTOS-EUROS is similar to DALES ($\sim$20 m within the ABL), terrain-following vertical layers are used in LOTOS-EUROS, whereas in DALES the surface is assumed to be flat to simplify the turbulence and thus topographical variations are not explicitly accounted for. Yet, for the large part of the Netherlands, the effect of topography is rather small. The $CO_2$ setup of LOTOS-EUROS uses emissions from the CoCO$_2$ project (https://coco2-project.eu/, last access: 27 November, 2024) similar to DALES except for the high-resolution emission disaggregation explained earlier in this section.

## 6 Results and validation

### 6.1 Comparison of simulation results

Figure 6 shows near-surface hourly-averaged $CO_2$ measured in parts per million (ppm), from DALES (upper panel) and LOTOS-EUROS (lower panel) for different times (6, 10, 16, 22 UTC) on the first day of the analysed period (June 26, 2018). As expected, DALES shows a more detailed representation of $CO_2$ sources and their transport across the region than LOTOS-EUROS. In the morning hours (6 UTC, Figure 6a), elevated mole fractions are observed along major roadways, caused by increased traffic emissions during morning rush hours, as well as in urban and industrial areas. The latter show up as bright plumes of elevated $CO_2$. This is due to the combination of stable thermodynamic conditions and shallow boundary layers. The southwestern part of the region shows a noticeable intensification of $CO_2$, likely due to point source industrial emissions, which dominate over other emission sources. Although LOTOS-EUROS reflects these emissions well, they appear to be more dispersed, with less recognizable differences between source types. Additionally, LOTOS-EUROS shows slightly higher near-surface $CO_2$ (up to 10 ppm more) than DALES around urban areas compared to DALES at 6 UTC. This might be explained by less vertical mixing in LOTOS-EUROS than in DALES at this height and the absence of a vertical component of emissions in LOTOS-EUROS.



**Figure 6.** Simulated near-surface (12.5m height) hourly-averaged $CO_2$ mole fraction (ppm) for different day times on 26-06-2018 (in UTC for the end of the averaging period). **(a)** 6 UTC, **(b)** 10 UTC, **(c)** 16 UTC; **(d)** 22 UTC. The domain covers the region from approximately 51.7° to 52.35°N and 4° to 6.0°E. Red stars mark the locations of observations (see Figure 5). Upper panel: DALES at 100m horizontal resolution; Lower panel: LOTOS-EUROS at ~2km horizontal resolution.



At local noon time (10 UTC, Figure 6b), both models show a reduction in $CO_2$ mole fraction across urban areas and emission plumes, consistent with enhanced atmospheric mixing as the boundary layer thickness increases. The $CO_2$ decrease of about 5-10 ppm over land is explained by biogenic $CO_2$ uptake through photosynthesis. Both models show similar trends in this reduction, though variations in spatial detail remain.

In the late afternoon (16 UTC, Figure 6c), there is greater spatial variability and a more pronounced decrease in background level, which is quite similar in both models ($\sim$15 ppm), although the overall range of $CO_2$ molar fractions remains similar to earlier in the day. DALES continues to show concentration signals that can more easily be attributed to local emissions, particularly along transportation routes associated with the evening traffic peak. LOTOS-EUROS captures these patterns, but presents them in a more smoothed manner due to its coarser resolution.

Around the local midnight (22 UTC, Figure 6d), the simulated $CO_2$ mole fraction distribution shows more stable conditions. Reduced atmospheric mixing at night leads to higher $CO_2$ mole fraction around urban areas. Yet, traffic emissions are much decreased at this time, as expected. Both models reflect this nocturnal pattern, though LOTOS-EUROS continues to show higher near-surface concentrations during nighttime, though less pronounced than seen for the morning hours.

Thus, diurnal variations in atmospheric $CO_2$ near the ground are represented generally well in both models, reflecting changes in background, anthropogenic emissions, and biogenic activity under varying atmospheric conditions. DALES provides a more detailed representation of individual emission sources and spatial variability, while LOTOS-EUROS, due to its coarser resolution, resolves the different source types less well. However, before it can be concluded that DALES provides a more accurate representation of $CO_2$, both models need to be compared against actual measurements, which we will turn to next.

## 6.2 The modelled $CO_2$ against Cabauw tower observations

The time series of atmospheric $CO_2$ mole fraction computed with DALES for the $CO_{2bg}$ and $CO_{2sum}$ tracers, are compared to LOTOS-EUROS as well as $CO_2$ measurements from Cabauw tower in Figure 7 for the period 26-28 of June 2018. Here, we show anomalies, after subtracting the $CO_2$ mean level over the considered period. This approach highlights the $CO_2$ variability relative to a baseline, emphasizing deviations from average conditions. The time series of the net ecosystem exchange (NEE) influence on $CO_2$ calculated from DALES has been added to show its contribution to the overall variability. To properly assess the capability of DALES to reproduce the observed variability during several consecutive daily cycles, all simulation results were linearly interpolated to the horizontal coordinates of the Cabauw tower using the air density to interpolate vertically.



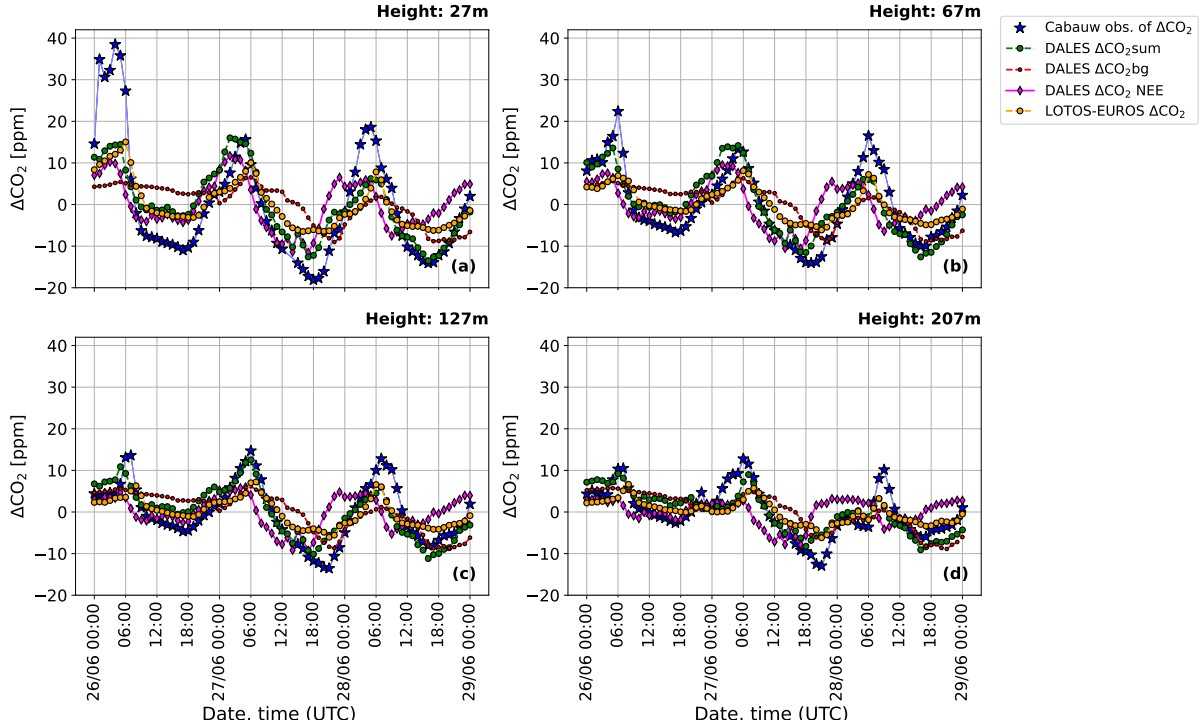

**Figure 7.** Time series of observed and modeled $CO_2$ mole fraction anomalies (ppm) at different levels of the Cabauw tower: **(a)** at 27 m; **(b)** at 67m; **(c)** at 127m; **(d)** at 207m. Blue stars: the anomalies of observed $CO_2$; green circles: the modeled $CO_2$ anomalies from DALES $CO_{2sum}$; red circles: the modeled $CO_2$ anomalies from DALES $CO_{2bg}$; purple diamonds: the modeled $CO_2$ NEE anomalies from DALES; orange circles: the modeled $CO_2$ anomalies from LOTOS-EUROS. All presented values are calculated by subtracting the $CO_2$ mean value (the average $CO_2$ mole fraction over the 26-28 of June 2018 period). The values are hourly-averaged, with time corresponding to the end of the averaging period.

During this period, the anthropogenic contribution was minimal due to the rural location and wind direction, resulting in a comparison of biogenic and background variability in the models and observations. Both DALES and LOTOS-EUROS capture the diurnal $CO_2$ cycle well, showing periodic variations that generally align with the observations, indicating a significant diurnal cycle in respiration and photosynthesis as expected in June. Daytime conditions exhibit more turbulence than nighttime, resulting in faster upward mixing and thus lower $CO_2$ values compared to the more stable nighttime hours, when vertical mixing is suppressed.

DALES $CO_{2sum}$ tends to follow daytime variations more closely than LOTOS-EUROS on June 27th and 28th, at lower levels (Figure 7 (a,b)), though showing comparable values during the start of the period. DALES simulated NEE explains some of the daytime $CO_2$ decline due to photosynthesis, which is captured better in DALES than LOTOS-EUROS. However, an underestimation of this decline remains, with observed $CO_2$ values being around ±5 ppm lower than in DALES and ±10 ppm in LOTOS-EUROS. This underestimation could be explained in part by an offset in the background level, especially in





times when the $CO_2$ loss due to photosynthesis declines (see red line in Figure 7). Errors in the background concentration could result from the coarse resolution of the original CAMS dataset as well as its 6-hour update frequency, which may not capture finer temporal variations. Besides, both models struggle to reproduce the observed nighttime $CO_2$ peaks. This issue is related to their inability to accurately simulate stable nocturnal boundary layers, where turbulence is minimal and stratification

predominates. This is especially evident at the start of the period at 27m height, where the deviation between modeled and observed values is as large as $\pm$ 25ppm. This is a common limitation of mesoscale models that LES does not solve yet (Umek et al., 2022).

At higher tower levels (Figure 7 (c,d)), the variability diminishes, which is consistent with the trapping of $CO_2$ in a shallow surface layer during night time, though the biases relative to observations persist. The overestimation of $CO_2$ molar fraction

during daytime, particularly in the late evening when photosynthesis declines, could be explained by the poorly resolved background.

Furthermore, biases in modeled wind speed and direction compared to observations could also contribute to discrepancies in $CO_2$ variability (Zheng et al., 2019). Offsets due to the absence of the vertical component of vegetation in the A-gs scheme of DALES should be minimal for the Cabauw location, as it is surrounded by grassland.

To further evaluate the results from DALES, we conducted a comprehensive statistical analysis. This analysis includes several metrics to quantify model performance, including linear regression, correlation, normalised standard deviation, the centred Root Mean Square Difference (RMSD) (after removing the mean bias between the modeled and observed data), as well as other statistical metrics such as Mean Bias Error (MBE) and Root Mean Square Error (RMSE). The results of the statistical analysis are presented in Figure 8 and in Table A3. The bootstrap analysis of mean absolute error has been also

performed and results presented in Figure C1 in Appendix.

For all presented heights at the Cabauw tower (left panel of Figure 8), the observed variability is better captured by $CO_{2sum}$ compared to $CO_{2bg}$, as indicated by approximately 1.5 times higher $R^2$ values ($R^2$: $\sim$0.7 versus $\sim$0.4). The predictions from both DALES and LOTOS-EUROS show similar performance in capturing the $CO_2$ variability in the measurements, with $R^2$ values slightly higher for LOTOS-EUROS compared to DALES $CO_{2sum}$ ($R^2$: $\sim$0.75 versus $\sim$0.7). With increasing height, the

performance of both DALES $CO_{2sum}$ and LOTOS-EUROS decreases, exhibiting larger biases against the observations at 207m height, with $R^2$ values of 0.63 and 0.71, respectively.

The statistical metrics presented in the Taylor diagram (right panel of Figure 8) illustrate the performance of different $CO_2$ simulations (DALES $CO_{2sum}$, DALES $CO_{2bg}$, and LOTOS-EUROS) in comparison to observations across various altitudes. The diagram shows that LOTOS-EUROS achieves a closer match to the observed variability in terms of correlation coefficients,

which are approximately 5% higher than those of DALES $CO_{2sum}$. However, other metrics favour DALES $CO_{2sum}$. Specifically, the normalized standard deviation of DALES $CO_{2sum}$ aligns more closely with the observed one across all heights, though this alignment is weakest at the lowest height (27m), where the observed magnitude of variability is larger (see Figure 7).

RMSD values for DALES $CO_{2sum}$ are slightly lower than those for LOTOS-EUROS across all heights, suggesting that DALES $CO_{2sum}$ captures the observed variability in magnitude slightly more accurately (RMSD: 27m: 8.28 vs 9.14; 67m: 4.59

vs 5.46; 127m: 3.96 vs 4.37; 207m: 3.57 vs 3.87).





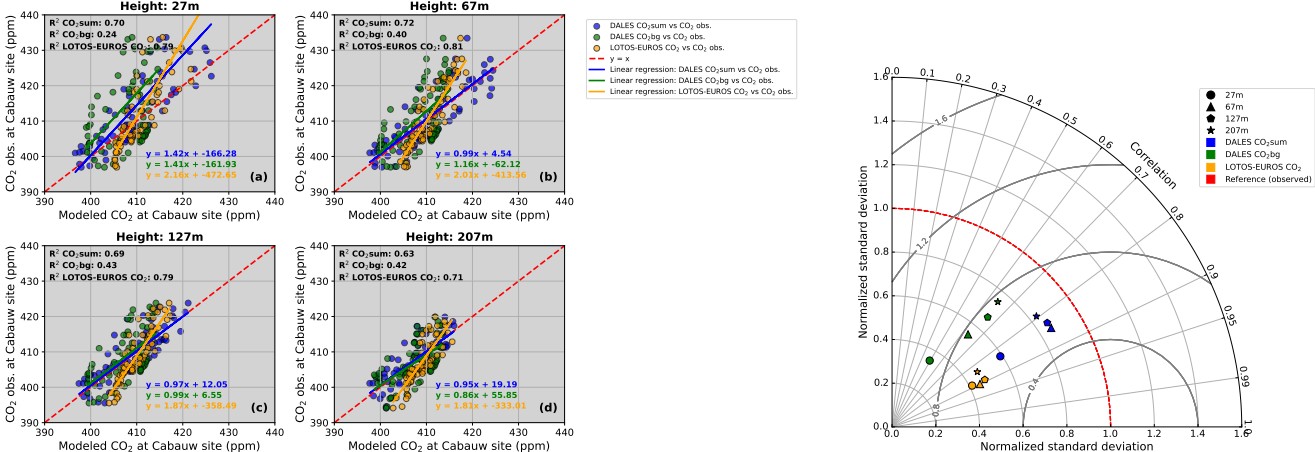

**Figure 8.** Left panel: Density plot comparing model predictions (DALES $CO_{2sum}$, $CO_{2bg}$, and LOTOS-EUROS $CO_2$) to observed $CO_2$ mole fractions (ppm) at the Cabauw tower for the period 26-28 of June 2018 at different heights: 27m **(a)**, 67m **(b)**, 127m **(c)**, and 207m **(d)**. The red dashed line represents the ideal relationship ($y = x$ line). Linear regression lines are shown for $CO_{2bg}$ (green), $CO_{2sum}$ (blue), and LOTOS-EUROS $CO_2$ (orange), along with the corresponding regression equations and $R^2$ values. Right panel: Taylor diagram quantifying the model performance against observations. Circle: 27m, Triangle: 67m, Pentagon: 127m, Star: 207m. Blue: DALES $CO_{2sum}$, Green: DALES $CO_{2bg}$, Orange: LOTOS-EUROS $CO_2$, Red: reference (observed $CO_2$). Grey circle lines represent values of RMSD.

In terms of MBE, DALES $CO_{2sum}$ generally exhibits larger errors compared to LOTOS-EUROS, except at the 207m altitude, where DALES $CO_{2sum}$ performs better (MBE: 27m: -4.63 vs -3.21; 67m: -0.53 vs -0.48; 127m: -0.28 vs 1.03; 207m: -0.25 vs 1.91). This indicates a slightly weaker performance of DALES $CO_{2sum}$ in predicting mean $CO_2$ levels for the first two heights and higher for the upper ones. Despite this, DALES $CO_{2sum}$ shows lowest RMSE values across all altitudes (RMSE: 27m: 9.49 vs 9.68; 67m: 4.62 vs 5.49; 127m: 3.97 vs 4.49; 207m: 3.58 vs 4.31).

Nevertheless, while there are subtle differences between the two models, the statistical metrics indicate that both DALES $CO_{2sum}$ and LOTOS-EUROS exhibit comparable performance at the rural Cabauw site, where both the local anthropogenic signal and spatial changes in $CO_2$ molar fraction remain relatively weak during the considered period, and finer resolution and explicit turbulence do not substantially contribute to increased predictive accuracy.

## 6.3 The modelled $CO_2$ against ground-based measurements in Westmaas and Slufter

A similar evaluation procedure has been applied for the Westmaas and Slufter measurement sites. The measurements at these sites were performed at one height (10 m), so model data are interpolated horizontally to the exact latitude and longitude of the measurements, but vertically the model data had to be extrapolated using the first two model layers, since the lowest model layer is slightly above 10 m. The time series of $CO_2$ mole fraction anomalies for the Westmaas and Slufter sites are presented in Figure 9.





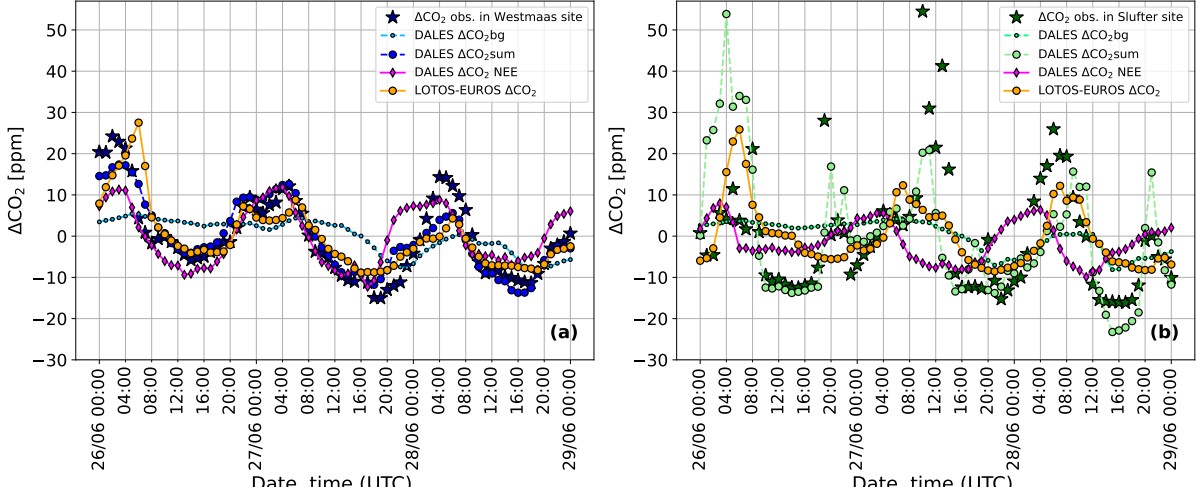

**Figure 9.** Time series of the observed and modeled near-surface atmospheric $CO_2$ mole fraction anomalies from Westmaas and Slufter at 10 m height for the period of June 26-28, 2018. **(a)** $CO_2$ mole fraction anomalies from Westmaas: observations (dark blue stars) and model predictions (DALES $CO_{2bg}$ in light blue dots, $CO_{2sum}$ in blue circles, DALES $CO_2$ NEE in purple, and LOTOS-EUROS $CO_2$ in orange). **(b)** $CO_2$ mole fraction anomalies from Slufter: observations (dark green stars) and model predictions (DALES $CO_{2bg}$ in light green dots, $CO_{2sum}$ in light green circles, DALES $CO_2$ NEE in purple, and LOTOS-EUROS $CO_2$ in orange). All presented values are calculated by subtracting the $CO_2$ mean value (the average $CO_2$ mole fraction over the 26-28 of June 2018 period). The values are hourly averaged, with time corresponding to the end of the averaging period.

At Westmaas (Figure 9a), the observed near-surface $CO_2$ (dark blue stars) shows typical diurnal variability, with lower $CO_2$ during the day due to vegetation uptake and enhanced vertical mixing, and higher concentrations at night due to stable ABL conditions and respiration. DALES $CO_{2sum}$ simulation (blue line) effectively captures the observed daytime declines, primarily due to the incorporation of $CO_2$ NEE in the simulations (see purple line). In contrast, the LOTOS-EUROS model (orange

line) tends to slightly overestimate $CO_2$ during these periods, which is also reflected in Figure 6a where the overestimation is well-pronounced, though its general pattern follows the observations. During nighttime, the model performance varies. On June 27th, DALES $CO_{2sum}$ matches observed concentrations relatively well, while LOTOS-EUROS deviates, predicting either consistently higher or lower values. However, on other nights, both models fail to accurately represent the nocturnal $CO_2$ accumulation, likely due to discussed limitations in nighttime stratification and mixing processes. This offset is particularly

noticeable in the early morning hours of 27th of June when modeled $CO_2$ values differ from the observations by up to 10-15 ppm.

In contrast, at the Slufter site (Figure 9b), both models show a greater disagreement with the observed $CO_2$ variability (dark green stars). Anthropogenic emissions dominate in the $CO_2$ variability in this location. DALES $CO_{2sum}$ reproduces some aspects of the nighttime variability and shows the tendency to reproduce the daytime variability, particularly $CO_2$ spikes

associated with localized anthropogenic emissions (see Figure 11), yet at lower accuracy. The NEE variability shows that



biogenic fluxes also play a significant role, especially during daytime. The LOTOS-EUROS model (orange) captures some of the observed pattern of nighttime variability well but fails to reproduce the finer-scale daytime spikes visible in the observations (see Figure 11).

Note that the Slufter site presents additional challenges to models due to its coastal location, where the interaction between
land, sea, and atmospheric dynamics introduces complex and unique $CO_2$ variability. These interactions, possibly involving sea breeze effects of temperature inversions, introduce fine-scale changes in $CO_2$ levels that might be difficult to reproduce even with the current 100 m resolution DALES setup. DALES does show a better daytime $CO_2$ variability than LOTOS-EUROS, pointing to the significance of local processes.

To further evaluate the accuracy of the simulations and quantify the degree of correspondence to measurements at the
Westmaas and Slufter sites, we performed the same statistical regression analysis as for Cabauw. The results of this analysis are shown in Figure 10.

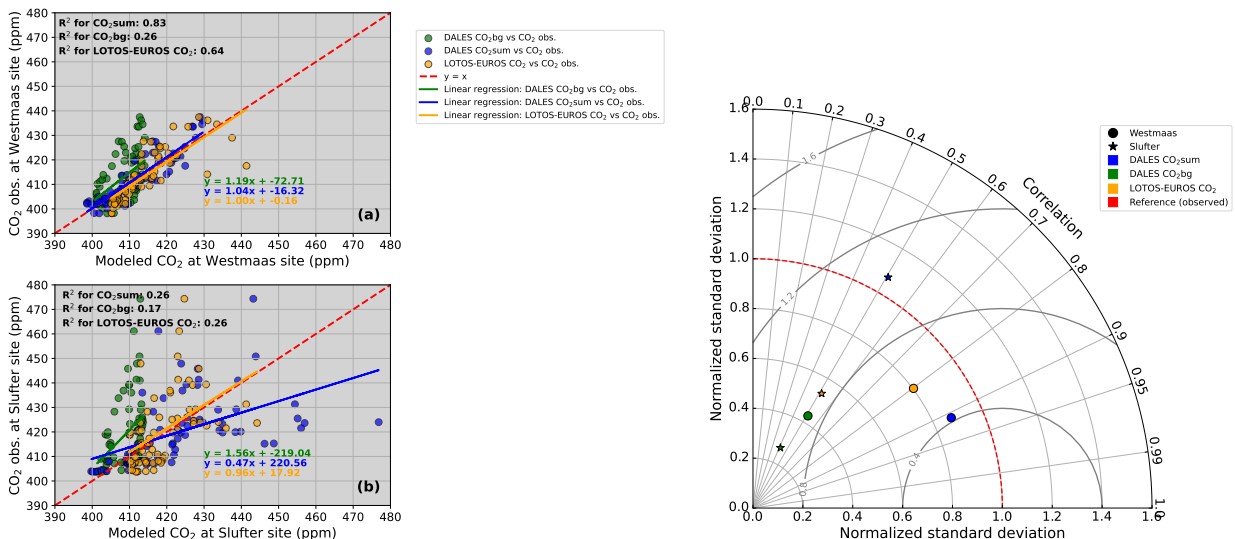

**Figure 10.** Left panel: Density plot comparing model predictions (DALES $CO_{2sum}$, $CO_{2bg}$, and LOTOS-EUROS $CO_2$) to observed $CO_2$ concentrations for the period 26-28 of June 2018 at Westmaas **(a)** and at Slufter **(b)**. The red dashed line represents the ideal relationship ($y = x$ line). Linear regression lines are shown for $CO_{2bg}$ (green), $CO_{2sum}$ (blue) and LOTOS-EUROS $CO_2$ (orange), along with the corresponding regression equations and $R^2$ values. Right panel: Taylor diagram quantifying the model performance against observations. Circle: Westmaas, Star: Slufter. Blue: DALES $CO_{2sum}$, Green: DALES $CO_{2bg}$, Orange: LOTOS-EUROS $CO_2$, Red: reference (observed $CO_2$). Grey circle lines are contours of equal RMSD.

At the urban background location of Westmaas (Figure 10a), the regression analysis reveals a significant improvement in model performance of the $CO_{2sum}$ tracer compared to $CO_{2bg}$. The $R^2$ value (0.83 vs 0.26), indicates that the surface fluxes in the model domain account for a substantial portion of the observed $CO_2$ concentration variability. The regression analysis also





indicates an improvement of $CO_2$ variability prediction using DALES $CO_{2sum}$ compared to LOTOS-EUROS ($R^2$: 0.83 vs 0.64) for this location.

    All statistical metrics derived from the Taylor diagram show an improvement in model predictions with DALES $CO_{2sum}$ compared to LOTOS-EUROS. This analysis shows higher correlation (0.91 vs 0.80), closer standard deviation to observed value (8.50 vs 7.81; observed std: 9.73), lower error metrics like RMSD (4.05 vs 5.81) as well as calculated MBE (-0.63 vs

0.90) and RMSE (4.13 vs 5.86), indicating lower overall errors and the highest accuracy of both variability and mean-level predictions in DALES $CO_{2sum}$ at this location.

    At Slufter site (Figure 10b), both models exhibit low $R^2$ values, with LOTOS-EUROS $CO_2$ showing an equal value to DALES $CO_{2sum}$ ($R^2$: 0.26) as well as equal correlation to observations (0.51 for both simulations). However, the normalized std DALES $CO_{2sum}$ is notably closer to the observed one than LOTOS-EUROS (15.27 vs 7.62; observed std: 14.23). RMSD is high,

suggesting that both models fail to the observed $CO_2$ concentrations and variability, albeit with slightly better predictions by LOTOS-EUROS (14.70 vs 12.21). Additionally, MBE and RMSE further highlight the slightly better performance of LOTOS-EUROS over the entire period (MBE: 3.14 vs -1.49; RMSE: 15.03 vs 12.30).

    Overall, this analysis underscores the limitations of the current model configurations, even using a high-resolution LES, in accurately capturing observed local $CO_2$ dynamics in complex and variable coastal environments as both DALES $CO_{2sum}$ and

LOTOS-EUROS show comparable challenges in reproducing $CO_2$ variability at Slufter.

### 6.4   Contribution of modeled local $CO_2$ components to regional $CO_2$ enhancement

One of the objectives of this study is to examine individual contributions of simulating the distinct compounds of atmospheric $CO_2$ to the total $CO_2$ that is observed at the measurement sites used in our study. To do this, we use the scalar $CO_2$ tracers in DALES from which the components of the atmospheric $CO_2$ can easily be determined (see Sect. 4.1). Figure 11 presents all

these components for each measurement site separately.





**Figure 11.** The hourly-averaged percentage contribution of various modeled local $CO_2$ components to the overall atmospheric $CO_2$ mole fraction [% of total] at three different measurement locations over the period from June 26 to June 28, 2018. Cabauw tower at two heights: **(a)** 27m and **(b)** 207m, and near-surface (10 m) at **(c)** Westmaas and **(b)** Slufter. The colored bars represent the contributions from three components of atmospheric $CO_2$: anthropogenic emissions (violet), $CO_2$ soil respiration (cyan), as well as $CO_2$ net assimilation (blue). The red dots represent $dCO_2$, which is the deviation of total $CO_2$ from the background [in ppm]. Each bar in the plots starts from 0, and to avoid overlap, the position of positive bars is adjusted such that bars with lower values are displayed in front. The values are hourly-averaged, with time corresponding to the end of the averaging period.

At the Cabauw tower location, a clear diurnal pattern in atmospheric $CO_2$ is observed at 27 m height (Figure 11a) and diminished at 207 m (Figure 11b) height. Here, $CO_2$ net assimilation has the largest contribution during daytime and soil respiration during nighttime (∼5% in total), reflecting the contribution of biogenic activity. At the higher altitude of 207 m, the diurnal pattern in both $CO_2$ biogenic components become smoother (∼2%), due to the increased distance from the surface. The contribution from anthropogenic emissions is visible but in general remains tiny especially at higher altitude (within 5 ppm).



As already discussed above, the absence of large urban areas nearby and the north-west wind direction during this period (see Figure 5) explain the low contribution of anthropogenic emissions to the $CO_2$ variability during this period. Thus, in this area the contribution of local agricultural emissions prevails, especially at the lowest tower level (see Figure 1).

At the Westmaas site (Figure 11c), we anticipated a stronger local $CO_2$ signal from anthropogenic emissions due to nearby
urban areas. However, $CO_2$ levels are only slightly above those measured at the lowest height at Cabauw (by 1-2%). This smaller difference may be due to the lower elevation at Westmaas (10 m vs. 27 m at Cabauw), combined with plume dilution and vertical mixing, which could significantly reduce the anthropogenic $CO_2$ reaching Westmaas. Pronounced variability in soil respiration contributes significantly to total $CO_2$ concentrations, particularly at night (up to ∼5%). Besides, as indicated in the $CO_2$ NEE time series in Figure 9, the negative contribution from the daytime photosynthesis $CO_2$ sink offsets the positive
contributions from both anthropogenic and soil respiration emissions, resulting in a net negative local contribution to the $CO_2$ concentration at day time throughout the studied period (by - 3%).

The Slufter site (Figure 11d) displays a distinct pattern in which anthropogenic emissions play a dominant role in the local $CO_2$ variability. When plumes from nearby facilities reach the measurement site the anthropogenic contribution exceeds 10% of the total $CO_2$ mole fraction. The influence of net $CO_2$ assimilation is also pronounced, though it is lower compared to the
Westmaas site. The $CO_2$ sink at Slufter is considerably weaker than at Westmaas, and it is insufficient to counterbalance the strong positive contributions from local anthropogenic activity, resulting in a strongly positive overall local contribution to the $CO_2$ variability at this location throughout the simulation period.

These results demonstrate the capability of the new DALES framework to support the assessment of local $CO_2$ sources and their contributions to atmospheric $CO_2$ concentrations. The weak signal of anthropogenic emissions at the Cabauw tower,
particularly at higher tower levels, contrasts sharply with the more urban or industrially influenced sites such as Slufter, where anthropogenic sources dominate. Besides, we show that biogenic $CO_2$ fluxes contribute significantly to diurnal variability. Even at Westmaas, at a short distance from the port and centre of Rotterdam, the biogenic component dominates the simulated diurnal $CO_2$ variability. This highlights the importance of an accurate representation of biogenic sources in high-resolution modeling of urban $CO_2$ variability.

## 7 Perspectives of LES development towards the simulation of $CO_2$ emissions

Despite significant progress in integrating $CO_2$ emissions into the LES model presented in this study, it is essential to address the existing and identified limitations and challenges to enable further improvements and more accurate future implementations.

In our work, we show limitations in LES in reproducing the observed variability, particularly under stable boundary layer conditions as well as in the coastal environment, as revealed in observations at Slufter. Although LES models still face chal-
lenges in accurately simulating nocturnal stable ABL conditions, proposed solutions have been developed to better resolve turbulence in stable boundary layers (de Roode et al., 2017; Dai et al., 2021). However, the verification of these methods is still ongoing, and the corresponding routines have yet to be implemented in the community version of DALES. The coastal environment is also an area for further improvement in LES, especially the integration of a more accurate marine atmosphere



in high-resolution models, which is planned to be done in the next few years within Ruisdael observatory. To achieve this will require an even finer spatial resolution to resolve the complex processes in the coastal environment. Yet, while DALES has the potential to operate at horizontal resolutions as fine as 1 m, the current 100 m resolution used in our setup is a compromise, balancing computational feasibility and domain size. These constraints are partly introduced by the meteorological data resolution from the HARMONIE-AROME model, which currently operates at ∼2.5 km resolution (N25 grid). However, from 2024 on, HARMONIE-AROME switches to the N20 grid, featuring finer horizontal resolution (∼1.3 km), which could enhance LES accuracy, particularly in regions with strong local $CO_2$ sources. Additionally, the 6-hour temporal resolution of the high-resolution CAMS reanalysis data used for $CO_2$ background levels introduces a limitation, especially when attempting to reproduce the observed diurnal variability. However, recent CAMS datasets offer the possibility to use 3-hourly data and higher horizontal resolution ($0.1° \times 0.1°$) (see https://ads.atmosphere.copernicus.eu/datasets/cams-global-greenhouse-gas-forecasts?tab=overview, last access: 27 November 2024), which could improve the representation of the $CO_2$ background level in future simulations. Also, as an alternative, LOTOS-EUROS output can provide better temporally resolved boundary conditions.

To increase the accuracy of LES simulated $CO_2$ variability, it is planned to switch from nudged to open boundary conditions. This has been shown to enhance the accuracy and applicability of the LES framework across diverse atmospheric conditions (Liqui Lung et al., 2024).

Besides, attention should be paid to the further improvement of anthropogenic emissions input. The national emission inventory for the Netherlands is continuously being updated, and data are expected to be more accurate in the future (Van der Net et al., 2024). In the meantime, ensemble experiments incorporating perturbations in anthropogenic emissions will be performed to address this uncertainty in modeling. The need for high-resolution activity and proxy data to advance the downscaling workflow and improve the emission input preparation for LES is also an important area for further work. Additional refinement could be achieved through the use of high-resolution monitoring data, such as ship traffic and waterway data, as well as detailed agricultural land use information across the Netherlands (van der Woude et al. (2023), https://www.clo.nl/en/indicators/en006111-land-use-in-the-netherlands-2015, last access: 27 November 2024).

Moreover, the vertical allocation of emissions and plume rise are important to be further improved. An alternative to the approach adopted here could be to address plume rise in LES by prescribing a heat source in the potential temperature equation at the location of the chimney top. This method enables LES to compute the heating tendency, which in turn modifies the vertical velocity through its effect on buoyancy. A key challenge is to accurately estimate the heat production at the stack, which represents the energy added by the emission of heated air per unit time. This would also help in representing the interaction of the emission plume with the ambient meteorological conditions. Note, that this works if the model grid is very fine (<50m), and that if the plume is narrow compared to the grid. Furthermore, our plume rise algorithm does not yet consider the influence of the emitting plume on the local meteorology, like thermal, radiative effects on atmospheric stability (Lohmann and Feichter, 2005).

The vertical allocation of emissions remains a complex challenge. In DALES the vertical profiles are not source-specific, but rather a simplified even distribution of the emissions between the calculated bottom and top of the plume for SNAP categories,



which include vertical component. As pointed out by Brunner et al. (2019), there are benefits of applying accurate category-specific emission initial vertical distribution.

As this study shows, the results are sensitive to the representation of biogenic $CO_2$ fluxes, even at short distances of urban centres. Hence, future efforts should also improve the representation of these fluxes at the resolution of the model, moving beyond our highly simplified split between grasslands and forest. In the case of intensive agriculture in the Netherlands this is complicated by a significant role of management. The measurement from the Loobos observation station made within Ruisdael Observatory can be helpful in this context as this station is in the forested area at distance from important anthropogenic

emissions, but with a region of intensive agriculture to the west. In addition, the incorporation of forested land may be upgraded to consider the vertical height of the forest, which also influences the simulated atmospheric dynamics. It may require a large upgrade of the model code, including the possible implementation of flexibility of trees representation if the model resolution changes. Yet, even much less sophisticated solutions could bring important improvements. The same holds for the representation of the urban landscapes. Currently, three-dimensional city maps for The Netherlands are under construction at

TU Delft (https://3d.bk.tudelft.nl/projects/, last access: 27 November 2024).

     Overall, ongoing research and development in these areas is needed to exploit the full potential of LES and increase the accuracy of modeling atmospheric $CO_2$ concentration variability.

## 8   Conclusions

We present a new atmospheric modelling platform for simulating the spatiotemporal $CO_2$ concentration variability at hectome-

ter resolution. The main novelty is to calculate the turbulent mixing and transport of $CO_2$ in the Dutch environment explicitly by means of the Dutch atmospheric LES. In this work, we present and discuss a workflow for downscaling the km-scale national emission inventory, consisting of point and area diffuse sources, into 100 m scale DALES input.

     We extended DALES with methodology to account for the vertical distribution of emissions from elevated point sources, including the modeling of plume rise. This is done using an online algorithm, which considers the interaction between plume

properties and the ambient atmospheric conditions. To represent biogenic $CO_2$ fluxes from respiration and photosynthesis, DALES has been extended with a simplified land surface model that differentiates between grassland and forest. The performance of DALES has been evaluated using the LOTOS-EUROS model and the available in-situ observations during a three day test period in June 2018. A rigorous statistical analysis quantifies the benefits of the high-resolution modeling approach, particularly near urban and industrial areas. This is evident in the standard deviations, which are closer to the observed values

across all measurement sites and generally lower RMSD. For instance, DALES $CO_2$sum at Slufter has a std of 15.27 ppm, closely matching the observed std of 14.23 ppm, compared to 7.62 ppm in LOTOS-EUROS. Additionally, DALES $CO_2$sum at Westmaas demonstrates improved performance with $R^2$ and correlation values of 0.83 and 0.91, respectively, alongside a lower RMSD of 4.05 ppm, compared to LOTOS-EUROS values of 0.64, 0.80, and 5.81 ppm, correspondingly (see Table A3 for more detail). Besides, we also identified limitations of the current framework, which require further improvement in future

development.



A multi-tracer approach is used to keep track of the contribution of different local sources to the simulated $CO_2$ concentrations. This analysis enhanced our understanding of the relative importance of anthropogenic emissions and biogenic fluxes in explaining the $CO_2$ variability in the measurements that have been used. The significant $CO_2$ concentration variations observed at Slufter on the Maasvlakte at the Western tip of the Rotterdam harbour is largely explained by anthropogenic activity (up to 10% of total $CO_2$ for the considered period). We also show the importance of the ecosystem fluxes of $CO_2$, even in close proximity to urban and industrial $CO_2$ emissions. These fluxes contribute largely to the $CO_2$ concentration variability especially during daytime when they may even cancel out anthropogenic concentration enhancements (as seen in Cabauw and Westmaas, where the daytime contribution of local $CO_2$ is negative, reaching -3%), emphasizing the importance of an accurate representation of biogenic processes in modelling urban $CO_2$.

The DALES framework has a significant potential to advance the modeling of atmospheric $CO_2$ concentration and support the independent evaluation of national emission inventories at the urban scale. This framework is expected to facilitate the quantification of local emission hotspots in combination with inversion techniques, while also reinforcing air quality monitoring efforts. Furthermore, by delivering detailed information on sub-scale processes, DALES can enhance parameterizations in larger-scale models through nesting, contributing to more accurate regional climate predictions (Sun, 2016). Ultimately, these advancements will support more informed decision-making and the formulation of effective policies aimed at mitigating climate change and its associated impacts, both in the Netherlands and beyond.

*Code and data availability.* The emission inventory used in this study were obtained from the Emission Registration (ER) portal, processed by the National Institute of Public Health and the Environment (RIVM) and are accessible at https://data.emissieregistratie.nl/export, last access: 27 November 2024. The Dutch Atmospheric Large-Eddy Simulation (DALES) 4.4 with emission module, developed in this study, is an open-source code available under the GNU GPL version 3. This specific version of the DALES model is available at Zenodo repository (Karagodin-Doyennel (2024a)). HARMONIE-AROMA model data on rectilinear grid, specifically the "Winds of the North Sea in 2050" (WINS50) dataset covering the Netherlands with a 1-hour temporal resolution, is available at https://dataplatform.knmi.nl/dataset/wins50-wfp-nl-ts-singlepoint-3, last access: 27 November 2024. CAMS (Copernicus Atmosphere Monitoring Service) data can be freely accessed at https://ads.atmosphere.copernicus.eu/, last access: 27 November 2024. The complete "offline" emission downscaling workflow program, developed and utilized in this study, is open-source and freely accessible code available at Karagodin-Doyennel (2024b). The CBS Vierkant 100 × 100 m and ESRI Shapefile datasets, which were used within downacaling procedure, are available at the CBS website: https://www.cbs.nl/nl-nl/dossier/nederland-regionaal/geografische-data/kaart-van-100-meter-bij-100-meter-met-statistieken, last access: 27 November 2024. Data from the Cabauw measurement site, used for validation, are accessible via the ICOS Carbon Portal: https://data.icos-cp.eu/portal/, last access: 27 November 2024.

**Appendix A**





**Table A1.** Classification of Anthropogenic Emission Sources by SNAP Category used in our study

| SNAP Category | Description | Comments |
|---|---|---|
| SNAP 1 | Power Generation | Refers to emissions from electricity generation facilities. |
| SNAP 2 | Residential and Commercial | Includes emissions from household and commercial heating/cooling. |
| SNAP 3 | Industrial Combustion | Emissions from combustion in industrial facilities. |
| SNAP 4 | Industrial Process | Emissions from industrial manufacturing processes. |
| SNAP 5 | Fossil Fuels Extraction and Distribution | Includes emissions from the extraction, processing, and distribution of coal, oil, and gas. |
| SNAP 7 | Traffic | Emissions from road transportation (passenger cars, trucks, etc.). |
| SNAP 8 | Other Mobile Sources | Emissions from non-road mobile machinery (construction equipment, ships, etc.). |
| SNAP 9 | Waste Treatment | Includes emissions from waste processing and treatment facilities. |
| SNAP 10 | Agriculture | Covers emissions from agricultural activities (livestock, fertilizers). |

**Table A2.** Parameters of the A-$g_s$ model used in DALES

| Symbol | Parameter | Value (grassland) | Value (forest) |
|---|---|---|---|
| $Q_{10,gm}$ | Temperature response coefficient to calculate gm [-] | 2.0 | 2.0 |
| $Q_{10,amax}$ | Temperature response coefficient to calculate Ammax [-] | 2.0 | 2.0 |
| $Q_{10,co2}$ | Temperature response coefficient to calculate the $CO_2$ compensation concentration [-] | 1.5 | 1.5 |
| $T_{1,gm}$ | Low reference temperature to calculate gm [K] | 278 | 278 |
| $T_{2,gm}$ | High reference temperature to calculate gm [K] | 301 | 305 |
| $T_{1,Ammax}$ | Low reference temperature to calculate Ammax [K] | 286 | 281 |
| $T_{2,Ammax}$ | High reference temperature to calculate Ammax [K] | 311 | 311 |
| $g_{min}$ | Cuticular (minimum) conductance to water vapor [m s$^{-1}$] | 2.5e-4 | 2.5e-4 |
| $a_d$ | Regression coefficient to calculate Cfrac [kPa$^{-1}$] | 0.07 | 0.07 |
| $K_x$ | Extinction coefficient of PAR inside the canopy [m ground m$^{-1}$ leaf] | 0.7 | 0.7 |
| $\alpha_0$ | Light use efficiency at low light conditions [mg J$^{-1}$] | 0.014 | 0.017 |
| $R_{10}$ | Respiration at 10°C | 0.23 | 0.1 |
| $g_{m,298}$ | Mesophyll conductance at 298 K [mm s$^{-1}$] | 7.0 | 3.0 |
| $A_{mmax, 298}$ | $CO_2$ maximal primary productivity at 298 K [m$^2$ leaf s$^{-1}$] | 1.7 | 2.2 |
| $f_0$ | Maximum value of Cfrac [-] | 0.85 | 0.89 |
| $CO_{2comp298}$ | $CO_2$ compensation concentration at 298 K [ppm] | 68.5 | 68.5 |



**Table A3.** Statistical metrics to evaluate the robustness of model performance against measurements at Cabauw, Westmaas, and Slufter locations

| Dataset | Location | Height (m) | $R^2$ | Correlation | Std (ppm) | RMSD (ppm) | MBE (ppm) | RMSE (ppm) |
|---------|----------|------------|-------|-------------|-----------|------------|-----------|------------|
| DALES $CO_2$sum | Cabauw | 27 | 0.7 | 0.84 | 8.17 | 8.28 | -4.63 | 9.49 |
| DALES $CO_2$bg | Cabauw | 27 | 0.24 | 0.49 | 4.82 | 12.18 | -7.04 | 14.07 |
| LOTOS-EUROS $CO_2$ | Cabauw | 27 | 0.79 | 0.89 | 5.70 | 9.14 | -3.21 | 9.68 |
| Ref. (observed) | Cabauw | 27 | N/A | N/A | 13.82 | N/A | N/A | N/A |
| DALES $CO_2$sum | Cabauw | 67 | 0.72 | 0.85 | 7.45 | 4.59 | -0.53 | 4.62 |
| DALES $CO_2$bg | Cabauw | 67 | 0.4 | 0.63 | 4.76 | 6.76 | -2.56 | 7.23 |
| LOTOS-EUROS $CO_2$ | Cabauw | 67 | 0.81 | 0.90 | 3.89 | 5.46 | -0.48 | 5.49 |
| Ref. (observed) | Cabauw | 67 | N/A | N/A | 8.69 | N/A | N/A | N/A |
| DALES $CO_2$sum | Cabauw | 127 | 0.69 | 0.83 | 6.07 | 3.96 | -0.28 | 3.97 |
| DALES $CO_2$bg | Cabauw | 127 | 0.43 | 0.66 | 4.73 | 5.35 | -1.12 | 5.47 |
| LOTOS-EUROS $CO_2$ | Cabauw | 127 | 0.79 | 0.89 | 3.38 | 4.37 | 1.03 | 4.49 |
| Ref. (observed) | Cabauw | 127 | N/A | N/A | 7.10 | N/A | N/A | N/A |
| DALES $CO_2$sum | Cabauw | 207 | 0.63 | 0.79 | 4.88 | 3.57 | -0.25 | 3.58 |
| DALES $CO_2$bg | Cabauw | 207 | 0.42 | 0.65 | 4.39 | 4.51 | 0.48 | 4.54 |
| LOTOS-EUROS $CO_2$ | Cabauw | 207 | 0.71 | 0.84 | 2.72 | 3.87 | 1.91 | 4.31 |
| Ref. (observed) | Cabauw | 207 | N/A | N/A | 5.86 | N/A | N/A | N/A |
| DALES $CO_2$sum | Westmaas | 10 | 0.83 | 0.91 | 8.50 | 4.05 | -0.63 | 4.13 |
| DALES $CO_2$bg | Westmaas | 10 | 0.26 | 0.51 | 4.19 | 8.39 | -4.53 | 9.55 |
| LOTOS-EUROS $CO_2$ | Westmaas | 10 | 0.64 | 0.80 | 7.81 | 5.81 | 0.90 | 5.86 |
| Ref. (observed) | Westmaas | 10 | N/A | N/A | 9.73 | N/A | N/A | N/A |
| DALES $CO_2$sum | Slufter | 10 | 0.26 | 0.51 | 15.27 | 14.70 | 3.14 | 15.03 |
| DALES $CO_2$bg | Slufter | 10 | 0.17 | 0.41 | 3.78 | 13.13 | -10.39 | 16.74 |
| LOTOS-EUROS $CO_2$ | Slufter | 10 | 0.26 | 0.51 | 7.62 | 12.21 | -1.49 | 12.30 |
| Ref. (observed) | Slufter | 10 | N/A | N/A | 14.23 | N/A | N/A | N/A |





## Appendix B

The expression B1 describes how to integrate point source emissions into the model for $CO_2$ tracer transport. The $CO_2$ tracer concentration is updated based on the point source emissions as follows:

$$CO2tracer_{izb} = CO2tracer_{izb} + \frac{point\_emis\_int_{izb}}{3600 \cdot \rho_{izb} \cdot dzf_{izb} \cdot dx \cdot dy \cdot 10^{-6}}, \quad \text{if } z_b = z_t, \tag{B1}$$

$$CO2tracer_{izb} = CO2tracer_{izb} + \frac{point\_emis\_int_{izb}}{2} \cdot \frac{1}{3600 \cdot \rho_{izb} \cdot dzf_{izb} \cdot dx \cdot dy \cdot 10^{-6}}, \quad \text{if } z_b - z_t = 1, \tag{B2}$$

$$CO2tracer_{izt} = CO2tracer_{izt} + \frac{point\_emis\_int_{izt}}{2} \cdot \frac{1}{3600 \cdot \rho_{izt} \cdot dzf_{izt} \cdot dx \cdot dy \cdot 10^{-6}}, \quad \text{if } z_b - z_t = 1, \tag{B3}$$

$$CO2tracer_{izb} = CO2tracer_{izb} + \frac{point\_emis\_bot_{izb}}{3600 \cdot \rho_{izb} \cdot dzf_{izb} \cdot dx \cdot dy \cdot 10^{-6}}, \quad \text{if } z_b - z_t > 1, \tag{B4}$$

$$CO2tracer_{izt} = CO2tracer_{izt} + \frac{point\_emis\_top_{izt}}{3600 \cdot \rho_{izt} \cdot dzf_{izt} \cdot dx \cdot dy \cdot 10^{-6}}, \quad \text{if } z_b - z_t > 1, \tag{B5}$$

$$CO2tracer_{izb+1:izt-1} = CO2tracer_{izb+1:izt-1} + \frac{point\_emis\_inbetween_{izb+1:izt-1}}{3600 \cdot \rho_{izb+1:izt-1} \cdot dzf_{izb+1:izt-1} \cdot dx \cdot dy \cdot 10^{-6}}, \quad \text{if } z_b - z_t > 1. \tag{B6}$$

where point_emis_int denotes the temporally interpolated point source emissions, obtained via the analogous interpolation method used for area emissions (see equation 2b); point_emis_bot = $(point\_emis\_int/(z_t - z_b + 1)) \cdot z_{b\_frac}$; point_emis_inbetween = $(point\_emis\_int - point\_emis\_bot - point\_emis\_top)/(z_t - z_b - 1)$ represents the emission distribution among layers between the bottom and top of the emission plume. This ensures mass conservation when $z_b - z_t > 1$. In the case of $z_b - z_t = 1$, the plume fraction factors are not applied the total emission is divided by 2 and equally distributed.





**Appendix C**

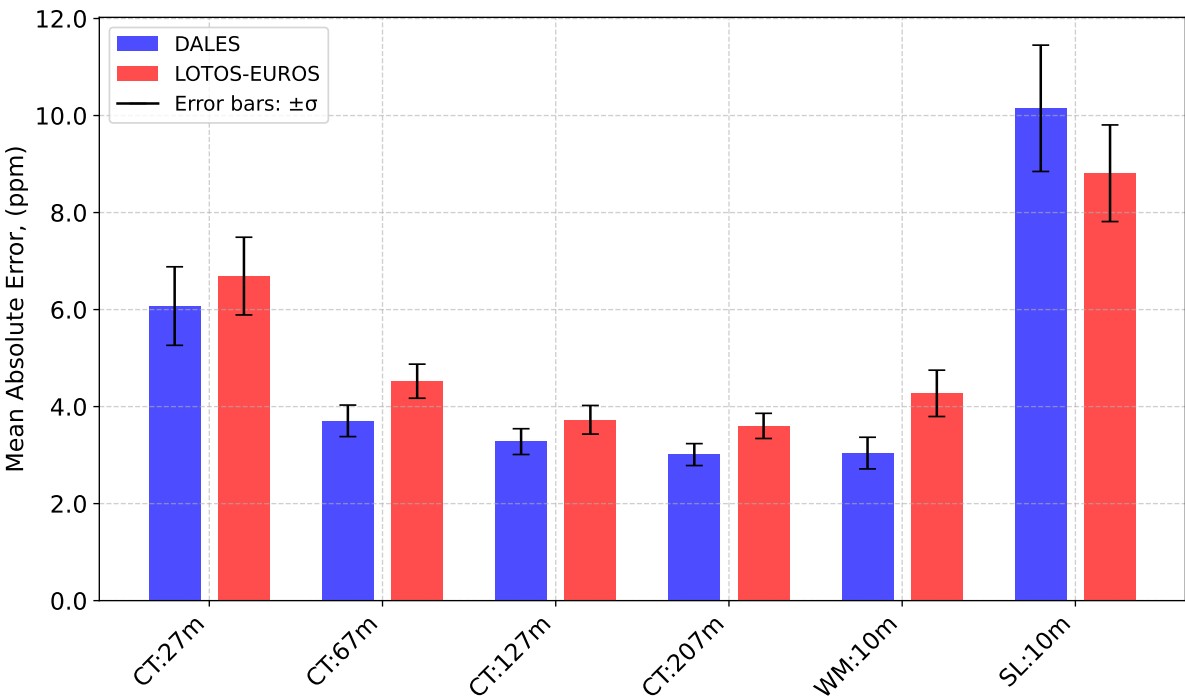

**Figure C1.** Bootstrap analysis (1000 iterations) of Mean Absolute Error (MAE) between model predictions (DALES $CO_{2sum}$ and LOTOS-EUROS $CO_2$) and $CO_2$ observations at multiple heights and locations. The bars represent the mean MAE, with error bars indicating $\pm\sigma$ based on bootstrap sampling. Locations and specific heights are labeled below the bars: CT:27m - Cabauw Tower at 27m; CT:67m - Cabauw Tower at 67m; CT:127m - Cabauw Tower at 127m; CT:207m - Cabauw Tower at 207m; WM:10m - Westmaas at 10m; SL:10m - Slufter at 10m.



*Author contributions.* A.-K.D. developed the emission downscaling workflow and DALES extentions, performed all simulations, handled the visualization and wrote the original draft with formal analysis. J.V.G.d.A. and B.v.S. assisted with the integration of forest components into the A-gs scheme and contributed to the results analysis. H.D.v.d.G. was responsible for the TNO observational data description and assisted with the analysis. F.J. supported the software implementation, initial process of data for assimilation, and assisted in the model
development. S.H. conceptualized the research work, developed the methodology, and contributed to the writing and analysis of the results. All the authors participated in editing the paper and discussing the results.

*Competing interests.* The authors declare that they have no conflict of interest.

*Acknowledgements.* We are grateful to the Dutch Research Council (NWO) for financial support of this research as part of the Ruisdael Observatory scientific research infrastructure (grant no. 184.034.015). We also extend our thanks to RIVM, especially to Margreet van Zan-
ten, Romuald te Molder, and Jolien van Huystee, for providing comprehensive emissions datasets, plume thermal properties, and assistance in their description and fruitful discussion. We thank Arjo Segers from TNO for providing LOTOS-EUROS simulation data and discussion on the setup. Additionally, we thank DAT.MOBILITY, particularly Eric Pijnappels, for granting access to the $NO_x$ traffic emission data used in this research. We are also grateful to ICOS for the opportunity to use $CO_2$ measurements from the Cabauw tower. Finally, we acknowledge SURFSARA for providing the computational resources to conduct the simulations for this study, and NWO for support-
ing the project budget through the National Roadmap for Large-Scale Research Facilities (https://www.nwo.nl/en/researchprogrammes/ national-roadmap-for-large-scale-research-facilities, last access: 27 November 2024).



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
