# Peer review of "Carbon dioxide plume dispersion simulated at hectometer scale using DALES: model formulation and observational evaluation"

_EGUsphere, 2024_

## Author Comment (AC1)

**Response to review 1**

Dear Referee,

Thank you for your detailed and constructive comments aimed at improving our manuscript. We appreciate the time and effort you have taken to provide thoughtful feedback. Below, we provide our detailed responses to your comments and suggestions to improve our manuscript (in blue):

1.  **The paper gets into too many very technical details about implementations, while some important characteristics of the simulations are missing. With a minor effort, one can calculate DALES domain is about 1100x1500 cells. How many layers are there, and what are they? How are the boundary conditions for mean wind, for turbulence, and for tracers implemented both in lateral and vertical directions? How much spin-up does the simulation need to become realistic?**

    **Response**: We thank the reviewer for their thoughtful feedback. We agree that providing more clarity on the simulation characteristics will enhance the understanding of our simulation setup.

    Indeed, the paper lacked information about the vertical grid structure. Therefore, we have added this information to the text (see L407-411): The vertical resolution ranges from approximately 20 meters (within the ABL) to a few hundred meters (in the upper troposphere). This is due to the use of a stretched vertical grid with 128 layers, an initial layer thickness of 25 meters ($dz_0 = 25$), and a stretching factor of 0.017 ($\alpha = 0.017$), which causes the layer thickness to increase geometrically with height.

    Regarding boundary conditions, periodic lateral boundaries are used in the DALES setup in our study. This periodicity applies to mean wind, turbulence, and tracers, meaning that what exits the east boundary re-enters at the west boundary and vice versa (the same applies to the north-south boundaries).

    In the vertical direction, the bottom boundary is a surface flux boundary, where fluxes are computed using the Land Surface Model (LSM) incorporated into DALES. This ensures that surface heterogeneity is properly accounted for in the simulation. The top boundary includes a damping sponge layer, which gradually reduces turbulence and tracers to minimize artificial reflections. These details have been added to the main text (see L199-202).

    Regarding spin-up and initialization periods that should be discarded from the analysis, after further discussion, we decided to adjust the analysis period. The periods to be discarded include not only the DALES initialization phase (i.e., the time required for fields to grow from small turbulent motions to fully developed turbulence), which typically lasts for the first 3 hours (e.g., https://gmd.copernicus.org/articles/12/5177/2019/gmd-12-5177-2019.pdf), but also periods with a stable atmospheric boundary layer (SABL) and HARMONIE forecast initialization.

The SABL (22-6 UTC) time period was excluded because DALES currently becomes less accurate when simulating the stable ABL at this 100 m horizontal resolution. At present, only simulations with horizontal resolutions finer than 10 m provide reasonable representations of the SABL (e.g., https://link.springer.com/article/10.1007/s10546-020-00558-1).

Additionally, the DALES meteorological boundary conditions are derived from the HARMONIE forecast, which resets daily as a new forecast is used. Therefore, when using the HARMONIE forecast for boundary conditions, it is generally advisable to exclude the first 4-6 hours from the analysis (https://gmd.copernicus.org/articles/17/2855/2024/gmd-17-2855-2024.pdf).

Based on this, our evaluation focuses on the diurnal period. We have decided to include June 25 in our analysis (which was previously fully excluded) while excluding the 23-6 UTC period from each simulation day to minimize errors caused by initialization and the well-known limitations of the current simulation approach. This has been described in the main text (see L420-430).

**I could not find how surface emissions are handled. Are they just injected into the lowest LES layer?**

**Response:** We thank the reviewer for this question. This information is provided in the main text (see L222-229 for area emissions and Sect. 4.1 for point sources).

We use different approaches to incorporate emission data into DALES. Since plume parameters for area emissions are not available and plume rise cannot be calculated, we compensate for this by following the methodology from Brunner (2019), which provides vertical distribution profiles for certain SNAP categories that are not emitted at the surface. For these categories, we apply emission profiles that redistribute surface emission values between ground level and a height of 150 m, as discussed in Brunner (2019). For other categories, such as SNAP 5, 7, and 10, emissions are injected into the lowest LES layer, as these categories correspond to surface emissions.

For point sources, we utilize available information on stack heights and plume parameters to calculate the vertical distribution (i.e., the plume bottom and top, between which emissions are injected) online using an algorithm based on Gordon et al. (2018) and Akingunola et al. (2018).

**There is little information on the reasons why specific technical decisions were taken. It is not clear why one would need to drive an ecosystem model for CO2 fluxes with high-resolution LES fields? Are the features such as second-scale updates of meteorology, clouds, physics of the ABL really needed to simulate CO2 fluxes by ecosystem, or are they just an overhead for such simulations? Same applies to plume-rise calculation.**

**Response**: We appreciate the reviewer for raising this important point. We want to clarify that we are not running an LES to provide details to an ecosystem model that we consider inherently necessary. Rather, we use the output of an ecosystem model in combination with an LES to reproduce $CO_2$ observations, ensuring consistency in

atmospheric CO₂ simulations when the ecosystem model is driven by high-resolution LES.

The decision to drive the ecosystem model for CO₂ fluxes in LES is motivated by the need to resolve fine-scale atmospheric processes that impact CO₂ dispersion and exchange. DALES incorporates a land surface model that simulates ecosystem fluxes (within the A-gs scheme), including CO₂ uptake through photosynthesis and release via soil respiration. Traditional mesoscale or global models often rely on parameterized boundary layer dynamics, which can introduce significant uncertainties in CO₂ flux estimations, particularly under heterogeneous land-cover conditions or complex meteorological scenarios. Additionally, features such as second-scale updates of meteorology and clouds are not merely an overhead but are essential for capturing rapid fluctuations in radiation and turbulence, both of which strongly affect photosynthesis and respiration rates.

These rapid fluctuations (from seconds to minutes) have been observed and lead to significant variations in heat, moisture, and CO₂ fluxes, as demonstrated in the figure below (Vilà-Guerau de Arellano et al., 2020):

[Figure]

**Figure 12.** IOP 2 (15 June 2018) time-series of (a) latent heat fluxes (LvE) at 1 min intervals with a displaced-beam laser scintillometer (DBLS) and at 10 min intervals with an eddy covariance system (EC) combined with scaled time series of photosynthetically active radiation (PAR, scaled by 1500 μmol m⁻² s⁻¹) and wind speed ($U$, scaled by 6 ms⁻¹). Panel (b) is the same as (a) but for the CO₂ flux ($F_{CO_2}$).

For example, cloud cover alters the amount of diffuse radiation, which in turn influences canopy light-use efficiency and CO₂ uptake. Similarly, resolving the sub-daily evolution of the boundary layer ensures a more realistic representation of plant stomatal responses and soil respiration fluxes, which are highly sensitive to temperature and moisture conditions. Thanks to new simulations that explicitly resolve clouds at high spatial and temporal resolution, we can improve key drivers of photosynthesis: direct/diffuse radiation, temperature, and water vapor deficit. We have briefly addressed the motivation for using an ecosystem model in LES in the introduction (see L64-70).

Regarding the plume-rise calculation for point sources, we use a parameterization approach because the spatial resolution of used LES grid (100 m) is not fine enough to explicitly resolve narrow plumes. As mentioned in the paper, explicit representation would require prescribing a heat source at the chimney top, allowing LES to compute

heating-induced buoyancy effects. However, this approach is only effective if the grid is very fine (<50 m) and the plume remains narrow compared to the grid. This is planned for future development. For our current simulation setup, parameterization remains the more suitable approach for efficiently representing plume rise above stack height when using hectometer-scale horizontal resolution.

Thus, in our case, incorporating an ecosystem model within DALES is not merely an added complexity but a necessary step to ensure a physically consistent representation of $CO_2$ exchange processes. Further elaboration is required to better understand how the main drivers of photosynthesis change under different conditions.

*Reference: Vilà-Guerau de Arellano, J., Ney, P., Hartogensis, O., de Boer, H., van Diepen, K., Emin, D., de Groot, G., Klosterhalfen, A., Langensiepen, M., Matveeva, M., Miranda-García, G., Moene, A. F., Rascher, U., Röckmann, T., Adnew, G., Brüggemann, N., Rothfuss, Y., and Graf, A.: CloudRoots: integration of advanced instrumental techniques and process modelling of sub-hourly and sub-kilometre land–atmosphere interactions, Biogeosciences, 17, 4375–4404, https://doi.org/10.5194/bg-17-4375-2020, 2020.*

2. **There is not much info on the computational costs of the DALES CO2 simulations. How many cores and hours needed to simulate one hour of the dispersion? How well it scales with number of cores? How much is the overhead for CO2 with respect to plain DALES? What are cost-benefits of using LES vs using e.g. LOTOS-EUROS at similar resolution for practical applications?**

**Response:** We thank the reviewer for this insightful question. Indeed, we did not include a detailed discussion of computational costs in the manuscript to not overweight it, and we appreciate the opportunity to clarify this aspect.

The computational cost of simulations with DALES depends, as in many other models, on multiple factors, including domain size, grid resolution, physics parameterizations, and general hardware performance (CPU or GPU using). In our study, we used the following setup:

Domain size: ~172.8 km × 115.2 km
Grid resolution: 100 m in the horizontal (1728 × 1152 grid points) and 128 levels in the vertical stretched grid (~20m resolution within ABL, and corses above, geometrically increasing).
Number of processes: 432 MPI tasks, with a decomposition of 18 × 24 processes
CO2 transport simulated using four scalar tracers (with different CO2 contributions as described in our paper).

Regarding performance, for our simulation setup and domain size, the wall-clock time for one hour of simulation is approximately 1.5-2 times real-time on our specified system and CPU-based simulation (GPU-based free version of DALES is in development). We did not explicitly measure the impact of additional scalar tracers on computational cost, but based on prior experience, adding four passive tracers to a model primarily increases memory usage and I/O operations rather than significantly affecting computational cost. Since these tracers do not involve additional chemistry or deposition processes (only ecosystem fluxes and plume rise for point sources) their

impact on the overall model dynamics is relatively modest. However, the increase in I/O and storage usage become a limiting factor, especially with high-frequency output, which is why the applied the limitation on the number of tracers to managing computational resources.

Concerning scalability, DALES is designed for distributed-memory parallelism using MPI. While strong scaling efficiency decreases at very high core counts due to communication overhead, DALES generally scales well up to several hundred cores for domains of this size.

The choice between LES (e.g., DALES) and an Eulerian chemistry-transport model like LOTOS-EUROS should depend on the research question and practical application. In addition, it is worth mentioning that LOTOS-EUROS cannot be run in LES mode. Therefore, the physics and dynamics in the model do not support achieving the same resolution as in DALES. In our case, we focus on the dynamics of $CO_2$ dispersion within the ABL and provide more reliable information of radiation, temperature and water vapor deficit fluctuations for the photosynthesis and soil moisture, making LES the preferred approach. Moreover, with using LSM with LES allows having more detailed soil temperature and moisture in the calculation of surface fluxes.

**Some of the used datasets have been declared public, but the procedure of obtaining them has not been clearly described. "The traffic data shapefile is provided by the Dat.mobility company and can be requested from RIVM." Publishing these assets via some open data publishing service or clearly identifying the specific dataset and providing a contact or URL would make it more clear.**

**Response**: We thank the reviewer for pointing this out. We agree that making the NOx traffic data available via Zenodo would improve accessibility. Therefore, we have included the shapefile in the Zenodo archive, along with the 'TVtot_N_sum' variable, which represents the total NOx emissions from all three vehicle types (light, medium, and heavy), and 'Shape_Length'. The dataset can be accessed here:
Doyennel, A. (2025). Annual NOx emission traffic shapefile (1.0) [Data set]. Zenodo. https://doi.org/10.5281/zenodo.14961517 We have added this link to the main text (see L383-384).

**The text is full of vague constructs that do not bring much information: "high-accuracy", "realistic approach", "fine resolution" etc. In a scientific paper each of such constructs if used has to have a very specific meaning.**

**Response**: We appreciate the reviewer's feedback and have carefully revised the text to replace vague or subjective terms such as "high-accuracy," "realistic approach," and "fine resolution" with precise quantitative values, comparative statements, or references to established standards (see e.g., L530-550 and L582-603).

**Specific comments:**

1. **Table 1 leaves a question on the quality of the fits used. It could be better replaced with regression plots used for get the fit. Then one could judge how good or bad the used approximation is. A simple three-panel figure would do: (a) log(flow rate) vs emission rate (of which species?); (b) Exhaust temperature vs emission rate for specific processes; (c) log height vs emission rate.**

   **Response**: We thank the reviewer for poining this out. We agree that the figure may be more representative than the table 1 in the paper to show the results of gap filling. Therefore, we replaced Table 1 in the paper with the figure presented below (see L308-309):

[Figure]

   The figure presents scatter plots of three plume parameters: volume flow rate (a), temperature (b), and stack height (c) as a function of emission rate (kg yr$^{-1}$). Each subplot combines the original data (blue stars) with the gap-filled values (red dots). A general regression fit (black dashed line) is also included in each panel.

   The volume flow rate (a) and stack height (c) show a generally positive relationship with emissions, captured by the regression trend whereas temperature (b) appears more scattered, with the regression fit suggesting a weak dependency on emission rate.

   Note that the resulting fit represents a general trend across the entire dataset, based on the relationship between the emission rate and the corresponding plume characteristics.
   It is important to mention that regression fitting is carried out separately for each subcategory of emission sources, such as chemistry, construction, oil, electricity, etc., with distinct fittings for each subcategory.

2. **Fig1 It might make sense to mention that SNAP8 includes shipping. SNAP10 has some marine emissions. What is the marine agriculture ?**

   **Response**: We thank the reviewer for pointing this out. Indeed, we agree that the term 'marine agriculture' does not exist. In our study, we categorized fisheries-related emissions under SNAP10 (Agriculture and Nature) rather than SNAP8 (which includes shipping). This explains the presence of some emissions in SNAP10 over the North Sea. We have updated Table 1 to explicitly mention this

classification in SNAPs used in our study (see L118-119). This choice was made because fisheries activities, including fuel use for fishing vessels and fish processing, might be more closely linked to the food production sector rather than transport activities. While these emissions could also be classified under SNAP8 due to the use of vessels (in the more recent emission inventories), the broader link to agriculture and resource extraction justified their inclusion in SNAP10. Some emission inventories (e.g., EMEP/EEA Air Pollutant Emission Inventory Guidebook) include fisheries emissions under agriculture and land use categories, particularly when they are reported together with other land-based food production emissions. Nethertheless, we recognize that classification choices sometimes may vary and appreciate your concern.

3. **Fig1 and Fig2 are labeled to have emissions in kg/year. Normally I would guess that it was meant kg/year/cell, however the figure has at least two distinct cell sizes, which is confusing. Should it be kg/year/ha then? It might make sense to mark the LES domain on these maps, to indicate that missing emissions from Germany (SE corner of the maps) do not affect the simulations.**

   **Response:** We appreciate the reviewer's comment. The emissions shown in the figures are indeed reprojected onto the target LES grid and are presented in kg/year per grid cell, with each grid cell being 100 × 100 m. The apparent differences in resolution arise because some categories undergo refinement, while others are simply reprojected; since the original resolution is coarse, the coarse grid structure remains visible. Additionally, over the ocean, the original resolution of emission inventory is 5 km instead of 1 km over land.

   We agree that specifying this more explicitly would improve clarity. Therefore, we have converted the emissions to units of kg/year per $m^2$. Additionally, we have zoomed out the map to show the full extent of the simulation domain and to clarify that emissions from Germany (in the southeastern corner) are absent in the simulation. However, emissions outside the domain should be accounted for in the CAMS EGG4 dataset, which was used for the $CO_2$ background concentration in DALES. The revised figure is provided below (see L350–351).

[Figure]

4. **l215. The shape file has not been specified properly.**

**Response:** We thank the reviewer for pointing this out. For refining, the road transport (SNAP 7) category, we use a NOx annual emission from the Dutch road network provided by RIVM. Specifically, we use a nationally comprehensive road network with attributes such as road segment length and NOx emission intensities from different vehicle categories: light vehicles (LVtot_N), medium vehicles (MVtot_N), and heavy vehicles (ZVtot_N) combined into 'TVtot_N_sum' (sum of NOx emissions from all these vehicle types). The shape file is placed to Zenodo archive and can be accessed here: Doyennel, A. (2025). Annual NOx emission traffic shape file (1.0) [Data set]. Zenodo. https://doi.org/10.5281/zenodo.14961517. We have added the link to this repository to the main text (see L383-384).

5. **l220 What is the advantage of downscaling 1km traffic emissions instead of just assigning CO2/NOx emission factors per vehicle type and use NOx inventory? At least the latter approach would give more consistent map.**

**Response:** Using high-resolution NOx traffic data for downscaling emissions allows for a more detailed and accurate representation of emissions at the local level. This method captures variations in traffic intensity and vehicle types, particularly in areas with heterogeneous traffic flows, which is critical for improving the precision of CO2 emissions in our model. In contrast, relying solely on emission factors and an NOx inventory might provide more uniform emissions map but lacks the granularity needed to represent local variations accurately.
Worth saying here is that emission factors can also be differentiated further within the modeling setup, but that the focus of our paper is on downscaling to LES resolutions.

6. **l260 What is the benefit of driving A-gs model with a few-second resolution? Is there any benefit in having it online with turbulence-resolving model?**

   **Response:** We thank the reviewer for this question. The benefit of driving the A-gs model with a few-second resolution lies in capturing rapid fluctuations in environmental conditions that directly affect plant physiological responses. For instance, as shown in Figures 8 and 9 from https://bg.copernicus.org/articles/21/2425/2024/, short-term variations in cloud cover, temperature, and vapor pressure deficit (VPD) lead to dynamic changes in net ecosystem exchange (NEE) and stomatal aperture. These rapid responses are typically averaged out in mesoscale models, which operate at coarser temporal resolutions and do not run in LES mode.

   By coupling the A-gs model online with a turbulence-resolving LES model, we can explicitly resolve sub-minute-scale interactions between the atmosphere and vegetation, rather than relying on parameterizations. This provides a more accurate representation of plant physiological responses under rapidly changing conditions, improving our understanding of carbon flux variability at fine spatial and temporal scales.

7. **l287 Is the model really able to conduct simulations and examine specific aspects of anything? I would say it should be a modeler..**

   **Response:** Agreed, we reconsidered this part (see L203-206).

8. **l290-310: Sec 3.1 describes unit conversion and linear interpolation in time. Both are quite trivial and, probably not worth the paper space. Same applies to appendix B.**

   **Response:** Agreed, we omitted this parts from the text (see L210-227).

9. **Sec 3.2 describes quite standard treatment of plume rise. A reference to Gordon (2018) and Akingunola (2018) would be sufficient to describe that, unless something different has been implemented in DALES. In the latter case a brief description with highlighting the differences would be needed.**

   **Response:** Agreed. We have significantly shortened this discussion, removing all expressions already presented in Gordon (2018) and Akingunola (2018) (see L229-255). However, we kept expressions 2 and 3, as they are not explicitly provided in Gordon (2018) and Akingunola (2018) in the form we presented them.

10. **Table A1 should have a proper reference.**

    **Response:** Agreed (see L112).

---

## Author Comment (AC2)

**Response to review 2**

Dear Referee,

Thank you for your detailed and constructive comments aimed at improving our manuscript. We appreciate the time and effort you have taken to provide thoughtful feedback. Below, we provide our detailed responses to your comments (in blue):

**The manuscript entitled: "Carbon dioxide plume dispersion simulated at hectometer scale using DALES: model formulation and observational evaluation" by Karagodin-Doyennel et al., is a comprehensive study in which CO2 mole fractions for a domain in The Netherlands (NL) were simulated using DALES, an LES model that runs at 100x100 m. Anthropogenic emissions (traffic, power, agriculture, residential, among others) and biogenic fluxes are added to the LES framework, so all flux components contributing to total CO2 mole fractions are taken care of. A specific processing pipeline is described for the downscaling of anthropogenic emissions to achieve the desired spatial scale. Finally, the CAMS air quality forecast is used as the CO2 background for the LES domain. The evaluation is done at 3 sites in the NL: Cabauw, Slufter and Westmaas using simulated tagged tracers to investigate the main drivers of variability at each site. There is quite some work on this study, the Figures are very well produced and overall the paper is well-written. However, the following are major points for improvement that the authors should consider before the article is accepted for publication:**

**General Comments**

**1. The article is presented with a lot of focus on anthropogenic emissions, which makes sense given the motivation in the Introduction hinting at the Paris agreement, emission reductions, etc. Therefore, the effort put on Section 2, preparing the SNAP categories, adjusting the point and diffuse area sources, downscaling emissions, plume rise, etc., is justified and well explained. However, in the Results from Figure 7 to 10, I was surprised that the authors do not have a single tracer with Anthropogenic emissions or even multiple tracers for each SNAP category. Instead, they show CO2bg, CO2sum, and CO2NEE with no opportunity to evaluate the individual contribution of the anthropogenic emissions. This is somewhat contradicting the set and setting given in the Introduction and all the effort done processing the emissions. Why not showing at least the CO2bg_emiss?**

**Response:** We appreciate the referee's comment and agree that a time series including all anthropogenic emissions is missing. To enhance clarity, we have now added this time series to better illustrate the contribution of anthropogenic emissions (See Figures 8 and 10).

However, our model setup does not include tracers for individual SNAP categories, as doing so would significantly increase computational demands for both simulation and

storage. While our current implementation does not feature a sector-specific breakdown, it can be adapted for future studies focusing on specific emission sources.

A detailed analysis of individual emission categories is beyond the scope of this work but is planned for a follow-up study. Therefore, in this study, we limit our focus to the combined emissions time series. The new figures are provided below:

[Figure]

The subtraction of the mean level over the period remains for observations, BG, and $CO_2$ sum tracers from DALES, as well as for LE $CO_2$ data. Additionally, we separately provide time series for NEE and anthropogenic $CO_2$ contributions (DALES $CO_2$ AE contr.).

**2. The use of CAMS as background is one of my major concerns. First, it is not entirely clear which CAMS product is used, the link provided ends up in the ADS website and from there several products providing CO2 mole fractions can be used, for example: CAMS global greenhouse gas forecasts, CAMS global**

greenhouse gas reanalysis (EGG4) or CAMS global inversion-optimised greenhouse gas fluxes and concentrations. Second,  as I suspect the forecast was used, for the modelling exercise the authors are performing they should show that the background does a good job in background stations and for this study in continental stations. From my experience at other latitudes, the CAMS forecast is biased low and likely would have an offset for your domain as well. I suggest to investigate this further and if needed bias correct the CAMS background used. The situation that the authors are simulating, with northeasterly winds will bring signals from the Groningen Gas Field, likely bringing $CO_2$  signals that will be present in the observations but not in CAMS (I think). You can, for example, compare CAMS $CO_2$ at Cabauw and at a background stations, like Mace Head (Ireland) and from there calculate a bias, if any, and correct your CO2bg tracer. But this analysis, leading or not to a bias correction, should be shown in Supplementary material.

**Response:** We thank the referee for valuable feedback on the use of CAMS data in our study. We acknowledge that there are several CAMS products providing $CO_2$ mole fractions with different quality, and it is important to clarify which product was used in our analysis.

The CAMS data we used are from the CAMS global greenhouse gas reanalysis (EGG4) product, which provides a refined, post-processed dataset that offers more accurate representation of greenhouse gases in the atmosphere with higher spatial resolution that other products. This product is based on delayed mode analysis, which ensures that some biases from real-time forecasts are corrected in the reanalysis version. We have added this to the main text (see L189-194)

Based on our experience and the geographic location of the study, we suspect there may be a low bias in the CAMS $CO_2$ predictions for the domain, as noted by the reviewer. Yet, as stated in the ECMWF report (https://confluence.ecmwf.int/display/CKB/CAMS%3A+Reanalysis+data+documentation#CAMS:Reanalysisdatadocumentation-CAMSglobalreanalysis(EAC4)Parameterlistings), the fossil $CO_2$ emissions  are included in EGG4. Thus, any effect from outside, like Groningen Gas Field emissions should be included in the CAMS data (Yet, based on previous assessments, the Groningen Gas Field has only a small contibution to fossil $CO_2$ emissions for recent years as the gas is not burned there but transported elsewhere).

Thus, the main concern that we mentioned in the paper, is the temporal resolution of the CAMS EGG4 dataset, which is 6 hours, which may cause significant errors at locations with an important contribution of the background to the local $CO_2$ variability (like at Cabauw). Besides, the horizontal resolution is also quite course (15 km) potentially affecting the background variability.  Yet, we agree that there are more refined CAMS products, e.g. with optimized fluxes, but unfortunately the spatiotemporal resolution of these data is low.

Concerning the identification of biases at backgound stations: for the CAMS global greenhouse gas reanalysis (EGG4) product this analysis has been done already by ECMWF (e.g., https://atmosphere.copernicus.eu/sites/default/files/publications/CAMS2_82_2023SC2_D82.4.2.2-2024_EGG4_2003_2023_v1.2.pdf). They report a positive bias at mid to high latitude stations like Alert and Barrow reaching up to 2.5% in recent years. This is assumed to be caused by underestimation of the carbon uptake in summer season. These biases are depicted in the Figure 3.1.1 and Figure 3.1.3 of the report. We have mentioned these biases in the discussion (see L534-538 and L593-597).

3. Length of article and structure. As I said before, there is a lot of work done here, but the reader probably does not want to read all about it, so I suggest to shorten the article and change the structure. In the Methods, I suggest to start with the DALES description and then continue with the emission preprocessing. In the Results, I strongly suggest to show the evaluation at the three sites all together, or start with Slufter and Westmass and then Cabauw. Now, the authors start by showing the weak part of the study: the comparison at Cabauw where LOTOS-EUROS performs better in most of the analysed statistcs. In addition, in each individual section of the Results the description of the plots are very subjective, using statements like: "typical diurnal variability", "slightly overestimate", "well-pronounced", "shows the tendency", "degree of correspondence", "lower accuracy", "levels that might be difficult to reproduce", etc. If you use the numbers you have at hand, this can be shortened significantly. One more thing, you devote quite some text to plume rise, but this is not discussed nor evaluated in the Results, consider shortening the plume rise part considerably or giving it more protagonism in your results.

**Response:** We agree that for Cabauw, LE represent more accurate results in term of correlation for all heights at Cabauw for this period. We think that larger descrepancies between DALES and observations are mostly because of the low resolution of the CAMS data. Nonetheless, in term of std, DALES does a very good job at Cabauw, showing very similar std's as observed. The location of Cabauw was mostly used for the testing the model, as in this work we mostly concentrated on urban area modeling.

Despite this, we agree, that it is logically better to first show the comparison with near-surface urban area observations at Westmaas and Slufter and after show background location of Cabauw, discussing current areas of limitations and improvement that can be done. Thus, we have switched these sections (see Sect. 8.2 and 8.3). Additionally, we reconsidered the results section and updated the text to reduce subjectivity (e.g., see L528-551; L583-603).
Plume rise has not been evaluated, which is not a trivial task to do, as there are no measurements of plume rise available. However, the original publications of the parameterization include a performance evaluation, quantifying its accuracy (Gordon et al. (2018), Akingunola et al. (2018)). We included the references to these papers (See Sect. 4.1).

4. Given that the authors include biogenic fluxes and that in the Results this signal is important for the interpretation, I think they should: i) explain with more detail how are

biogenic fluxes simulated or given as offline input to LOTOS-EUROS, ii) present the spatial distribution of grasslands and forests in DALES iii) if possible and the models permit it, evaluate the DALES/LOTOS-EUROS NEE fluxes at a flux tower within the domain.

**Response**: we agree that in the LOTOS-EUROS description, the explanation of how biogenic fluxes are treated is not given. In LOTOS-EUROS, $CO_2$ biogenic fluxes, including gross primary production (GPP) ($CO_2$ uptake by photosynthesis) and ecosystem respiration ($CO_2$ release from the soil and plants), are not simulated internally and handled using external datasets. LOTOS-EUROS data which we used in our analysis were taken from the CoCO2 project. Here, LOTOS-EUROS incorporates biogenic fluxes from the VPRM (Vegetation Photosynthesis and Respiration Model) flux data from DLR (Deutsches Zentrum für Luft- und Raumfahrt, the German Aerospace Center), as offline input to account for the biospheric contribution to $CO_2$ concentrations. The input and VPRM description can be found here in chapter 5 of Denier van der Gon. H.A.C et al, Deliverable report D2.2 Prior data 2021 documentation report, CoCO2 project: https://coco2-project.eu/node/365. We have added this to the main text (see L467-471).

As suggested, we have added a figure showing the spatial distribution of grasslands and forests used by the Ags scheme in DALES (see L175-177):

[Figure]

A direct comparison between NEE from LOTOS-EUROS and DALES is not possible in the current setup. LOTOS-EUROS does not provide the NEE contribution to $CO_2$ or biogenic fluxes as separate outputs. Similarly, DALES does not output biogenic fluxes separately. The NEE presented in the paper represents the NEE contribution to $CO_2$ mole fractions rather than the actual fluxes. Further validation of NEE and individual biospheric flux components in DALES is planned for a follow up study, as it requires modifications to the setup and output procedure.

5. As a GMD paper and given that one of the objectives of the paper is to "Document" the downscaling of the emission inventory, I think the authors should provide clear links to function names and modules in the code and give an overall idea of how the

preprocessor was constructed. For example in line 141, the authors write: "The workflow is structured as a comprehensive program with several stand-alone modules, each responsible for different aspects of emission data processing". So I suggest you present an overview of the program and those modules that you are referring to.

**Response**: Yes, we agree that providing an explanation of the modules would improve clarity. However, we chose not to include this discussion in detail to keep the paper concise and focused on the key findings. The full description of all modules is available in the GitHub repository (https://github.com/ruisdael-observatory/ghg_emission_inventory_workflow). Based on this, we decided to add names of modules in the main text (e.g., L302) but refering to the metadata information described in the GitHub documentation of the workflow. In the text, we have also included the link to the repository at the beginning of the section to ensure accessibility to the full program (see L300-301).

Specific comments:

L124: define E-PRTR, not done previously

**Response:** The E-PRTR (European Pollutant Release and Transfer Register) is a publicly accessible database that provides information on pollutant emissions from industrial facilities across Europe. We have now included its definition in the text (see L132–133).

L134: the 100m spatial res. is for tracer transport but not for resolving turbulence? not clear since model is not described first.

**Response:** The 100 m spatial resolution refers to the LES grid used in our study. This resolution is applied both for tracer transport and for resolving turbulence.

L148-149: "1316 out of 1914" add a percentage here.

**Response:** Agreed. We have added that ~68% of the data contain gaps in plume parameters and have specified this in the main text (see L307-308).

L156: I suggest to merge this two lines with the previous paragraph. In addition, what do you mean with "cohesive"?

**Response**: We agree and merged these lines with the previous paragrath. The word "cohesive" in this context means "complete". For clearty, we have replaced "insufficiently cohesive data" with "incomplete data" (see L314).

L165: "In essence" may be "originally" fits better?

**Response**: Agreed. We have replaced "In essence" with "originally" in the main text (see L323).

L167: Can you show an example of this separation on a Map? It will be informative to show the reader what is the output of this. Sufficient as SupFigure.

**Response:** We appreciate the reviewer's suggestion and agree that visualizing this separation would be informative. However, at the LES grid resolution of 100m, individual point sources are barely distinguishable (see the figure below, which shows only point sources).

Instead of adding an additional figure, we have expanded the description in the text to highlight that the main clusters of point sources within our domain are concentrated around Amsterdam (including the port of IJmuiden) and Rotterdam, particularly along the harbor and port zones. Additionally, we now specify that the ratio of point source to area emission contributions to total $CO_2$ emissions in the simulation domain is approximately 55%/45%, with point sources being the dominant contributor, as expected.

This discussion has been added to the main text (see L341-348). The figure below illustrates these clusters with red rectangles:

[Figure]

L169: "gpkg" is this a specific format in a given programming language or platform, specify.

**Response**: GeoPackage (GPKG) is an open, standards-based format for geospatial data defined by the Open Geospatial Consortium (OGC). It is a lightweight, platform-independent SQLite database designed to store vector features, raster maps, and related metadata in a single file. Using this format enables more accurate and efficient geospatial data processing by ensuring consistent spatial referencing, minimizing data loss during format conversions, and supporting complex spatial queries directly within the database. In our program, the use of the GPKG format was adapted from QGIS code to Python to enable more precise processing of spatial data. GeoPackage format has been specified in the main text (see L329-331).

L175: What about mass conservation in this procedure? Did you check that?

**Response**: yes, the mass conservation has been assessed and the difference between original data and processed is within permissible error (<5%). We make use of a mass fixer.

L176: This line reads like a caption.. I suggest removing it and start with the next paragraph.I dont think you need this sentence here.

**Response**: Agreed. We have removed this line and started with the next paragrath as suggested (see L337).

Figure 1. Can you point the reader to what do you consider as point sources and area emissions on the Figure? For example, for SNAP 2: Res. and Commercial. Are these considered point sources or area emissions? SNAP 5 is quite vague: Fossil Fuels? most of these can fall in SNAP 5. Given the level of detail you are aiming, I suggest to be more precise here. These SNAP categories represent individual tracers in DALES? Not clear.

**Response**: No, in DALES, we do not currently have sectorial split in the setup and individual tracers for each category of anthropogenic emissions, as analyzing individual sources is beyond the scope of this study and would require additional investigation. Instead, DALES incorporates the sum of anthropogenic emissions from all SNAP categories. However, we agree that it would be clearer to specify which of the presented emissions are considered point sources and which are classified as area emissions.
We have done it in the text, desciribing areas where the majority of point sources are located.
In the text we added: It is important to mention that the Rotterdam harbor, including its port infrastructure, and the Amsterdam area, together with the IJmuiden port, have the highest density of point source emissions within the simulation domain. The majority of point sources fall into the SNAP1 (power generation) and SNAP3 (industrial combustion) categories, as these sectors rely on large stationary facilities, such as power plants, refineries, and industrial manufacturing sites. The point source/area emission contribution ratio to the total $CO_2$ emissions at the simulation domain is ~55%/45%, with point sources having the larger contribution, as expected (see L343-349).
Also, we agree that it is not clear what emissions are included in SNAP 5, as in the figure this category named as "Fossil Fuel". In the Table 1, it is more clearer explained. SNAP 5 category is oil/gas extraction and distribution emissions, so it includes emissions from the extraction, processing, and distribution of oil, and gas. We have corrected this in the figure to make it more clearer for readers (see Table 1 and Figure 4).

L187: "responsible for the majority" how much, add %.

**Response**: from our estimation, this represents approximately 70% of total national anthropogenic emissions in the Netherlands (109 Mt/year out of ~160 Mt/year for 2018). We have included this information in the main text (see L355).

L188-190: This is not clear. Are there less sources in the northeast? Why only for evaluation of the model? Are you only simulating when the wind comes from the NE? Please clarify.

**Response**: The simulation can be performed for any domain within the Netherlands. However, for model evaluation, we selected a specific analysis period based on meteorological conditions and wind direction. This choice aimed to minimize the influence of emissions from neighboring countries, such as Germany, which are not included in the setup, ensuring a more controlled comparison.

Figure 2. I suggest to have Fig 1 and Fig 2 in one panel together. You can have Fig 2 as a large panel above the small panels of Fig 1. This will improve the visualization. "These emissions maps" should be singular instead?

**Response:** Agreed. We have merged these figures together in Figure 4 (see L350-351):

[Figure]

L194: So, in the model the plume rise mechanism is always activated, but for area emissions the bottom and top are set to fixed values, right?What is the interaction between plume rise and "natural buoyancy"? As I think of plume rise in emissions, their rising is due to temp gradients of the emissions and surrounding air. But this is not the case for area emissions… clarify please.

**Response:** Yes, that is correct. The parameterization of plume rise for point sources is based on the difference between the plume's characteristics and the surrounding meteorological parameters, which determines the residual buoyancy. This buoyancy is then used to estimate the plume rise height (the height where residual buoyancy = 0). For area emissions, we cannot apply the same parameterization since we lack specific plume characteristics. In most cases, area emissions are released at the surface (the first LES layer). However, for certain categories with a vertical component, such as SNAP 8 (where ship stacks contribute to emission distribution), we apply a fixed plume top of 150 m, based on Brunner et al. (2019). Therefore, in the current setup, plume rise for area

emissions does not depend on buoyancy. This explanation has now been included in lines L222-235.

L196-197: This contradicts the sentence in line 193 and 194. I suggest you describe the plume rise for point sources first and then for area emissions. It is confusing as it is now.

**Response:** We agree with the reviewer's suggestion. Both L193-194 and L196-197 discuss the vertical distribution of area emissions. Specifically, L196-197 clarifies that area emissions from SNAP 5, 7, and 10 are applied at the first LES layer, as they are surface emissions, while emissions from other SNAP categories are redistributed between 0 and 150 m. To improve clarity, we have reordered the text to better distinguish the plume rise applied for area emissions and point sources. This clarification can be found in lines L222-235.

L200: The temporal disaggregation is for all categories? only for traffic? do all have a diurnal cycle or rush hour? I dont think so. How do you deal with Agriculture and Industrial processes for example?

**Response:** Temporal disaggregation is applied individually for each emission SNAP category based on EDGAR coefficients, which are derived from the long-term statistics on the temporal variability of various emissions. For example, for SNAP 7 (traffic emissions), the coefficients account for variations between rush hours and calm traffic hours, as well as diurnal, weekly (workday/weekend), and seasonal cycles. However, not all categories follow a clear diurnal cycle. For categories such as agriculture and industrial processes, temporal disaggregation is based on general temporal emission patterns rather than specific rush hour variations. The details of how EDGAR temporal profiles have been obtained can be found in TNO (2011) and Crippa et al. (2020).

*TNO: Description of current temporal emission patterns and sensitivity of predicted AQ for temporal emission patterns, Tech. rep., EU FP7 MACC deliverable report DD-EMIS1.3, TNO, Princetonlaan 6, 3584 CB Utrecht, The Netherlands, https://atmosphere.copernicus.eu/sites/default/files/2019-07/MACC_TNO_del_1_3_v2.pdf, hugo Denier van der Gon, Carlijn Hendriks, Jeroen Kuenen, Arjo Segers, Antoon Visschedijk. For: EU FP7 MACC (Monitoring Atmospheric Composition and Climate), Grant agreement no.: 218793; coordinator: Dr. Adrian Simmons, ECMWF, UK, 2011.*

*Crippa, M., Solazzo, E., and Huang, G., e. a.: High resolution temporal profiles in the Emissions Database for Global Atmospheric Research,840 Scientific Data, 7, 121, https://doi.org/10.1038/s41597-020-0462-2, 2020.*

L202-203: you refer several times to the model description on subsequent section. I suggest you describe DALES first and then its inputs (emissions).

**Response**: We agree with the reviewer's suggestion. To improve clarity, we have reordered the sections to first describe the DALES model and its emission module (see Sect. 3) before introducing the downscaling workflow (see Sect. 5). Additionally, we now discuss the point source plume rise mechanism earlier in the text (see Sect. 4).

L210-211: "Data have 100 × 100 m2 resolution are freely available from the CBS website" something wrong in this sentence: "Data with 100 x 100 m2 resolution are freely...."

**Response**: Agreed. We have corrected the sentence to: "Data with 100 × 100 m$^2$ resolution are freely available from the CBS website." (See L374).

L222: "target level" add "target level (100m)"

**Response**: Agreed. We have added this (see L384).

L225-230: this sounds more like Discussion.

**Response**: Agreed. We have kept this section but shortened it and removed the final paragraph to maintain focus and avoid it sounding too much like a discussion. (see L387-391).

L231-234. With Figure 3 I am convinced, no need to state this in the Methods. In addition, so far I have not seen any reference to the downscaling code or repository.

**Rsponse**: Agreed, we removed the last paragrath of this section (see L391).

L255: What about autotrophic respiration

**Response:** Yes, the Ags scheme includes autotrophic respiration, which is the respiration of plants - which dominates in the NEE flux during night time. It should depend on a constant R10 (see https://link.springer.com/article/10.1007/s10584-006-9182-7, Equation 12). For clarity, we have specified this in the main text (see L158).

L271-272: Do you also need a high resolution map of the forests in NL? Im thinking about the big parks within cities and off course the Veluwe.

**Response:** Yes, the land surface model (LSM) input includes a field that distinguishes between two ecosystem types: grass and forest. This allows the model to identify grid points covered by grass and those covered by forests, including large parks within cities. A map illustrating this distinction has been added to the main text (see Figure 1 in L176-177).

L286: van Genuchten? what is that?

**Response:** Van Genuchten model parameters are used to describe soil water retention and hydraulic properties (see Vilà-Guerau de Arellano et al. (2015)). This explanation has been added to the main text (see L195-198).

Vilà-Guerau de Arellano, J., van Heerwaarden, C., van Stratum, B., and van den Dries, K.: Atmospheric Boundary Layer: Integrating Air Chemistry and Land Interactions, Cambridge University Press, 2015.

L294: which workflow? the previously described emission downscaling pipeline? Be specific here.

**Response**: Agreed. We have excluded this sentence from the main text as we have reordered the sections (see L210-215).

L301-302: How many vertical layers does the model have?

**Response:** In this study, the model uses 128 vertical layers, extending from the ground to approximately 6 km. The reviewer may be referring to $K_{emiss}$, which represents the highest layer where emissions are applied: LES layer corresponds to 150 m for area emissions and the plume top height (calculated interactively) for point sources. Netherthless, we have added the number of vertical layers used in DALES to the setup description for clarity (see L411).

L310: you meant point sources?

**Response:** Yes, by 'individual sources' we mean point sources. We have clarified this in the main text (see L229).

L316: this line is not needed I think.

**Response:** Agreed. We have excluded it.

L331: why not just say negative...?

**Response**: Yes, agreed. We also decided to shorten this part, as the plume rise mechanism is already described in the reference papers (see Sect. 4.1).

L341: u_zhj+1 and uzhj are the wind speed using the three components of the wind ? u, v and w?
**Response:** Thank you for pointing this out. Indeed, it was not specified but is mentioned in the reference paper (Gordon et al., 2018). In the parameterization, only the total horizontal wind speed is considered, so U is calculated as sqrt($u^2$ + $v^2$) averaged over the layer. The w-component is not considered. We have now clarified this in the main text (see L251-254).

L394-395: Well. I don't understand why you took a lot of effort defining point sources, area sources, downscale, regrid, etc, to have all as one tracer in DALES. I really was expecting to see tracers per SNAP category. This makes me think, why all the effort of Sections 2.1 and 2.2 to group all anthropogenic tracers together! Now the way you are setting up the experiments suggest you are more interested in the biogenic signals.

**Response**: As discussed earlier, we agree that having individual tracers for each SNAP category would be beneficial for analyzing their respective contributions. However, this level of detail goes beyond the scope of the current study and would be more suitable for a separate investigation.

That said, we disagree with the notion that processing point sources, area emissions, and individual SNAP categories separately while combining all processed emissions into a single tracer is meaningless. All the details discussed in Sections 2.1 and 2.2 are fully accounted for in this single tracer, as they all contribute to the quality of the overall emission component as well as the separation between the aggregated anthropogenic and biogenic contributions to CO2.

In this work, our goal is to assess the ability of a high-resolution (LES) model to reproduce CO2 measurements in the Netherlands. In addition, we aim to determine what fraction of the observed signal is due to anthropogenic vs. biogenic contributions. To achieve this, we need the spatial detail of DALES to ensure accuracy. Specifying the anthropogenic signal into its different components would be a next step, which the current setup allows, but this is not the focus of this paper.

Regarding biogenic fluxes, we acknowledge that CO2 exchange is a crucial part of the carbon cycle, as demonstrated in the results section. Thus, to improve clarity and show its contribution, we have included the $CO_2$ anthropogenic emission tracer in Figures 8 and 10 (See L529-530 and L584-585).

L404-405: Why? this is also mentioned before. Be specific why this is the case. Not clear.

**Response**: Overall, these conditions allow separating the contributions of anthropogenic and biogenic $CO_2$ sources by providing a stable meteorological environment with minimal interference from the strong winds or turbulent mixing which are good for the validation. The stable summer conditions, characterized by clear skies and relatively warm temperatures, led to pronounced diurnal variations in the boundary layer height. During early morning hours, weak atmospheric mixing resulted in the accumulation of $CO_2$ near the surface, enhancing the detectability of both anthropogenic emissions and biogenic fluxes.

L406: In other words, you do like a 1-day spinup?

**Response**: Yes, but upon reconsideration, excluding the entire day was decided to be an exaggeration. Therefore, we changed the period by including the June 25th in the analysis but excluding the period from 23:00 to 06:00 UTC of each day during the considered period. This is acceptable as DALES is influenced not only by the DALES initialization (that is first few hours) but also by the HARMONIE spin-up and inaccurate simulation of the night-time stable boundary layer conditions at 100m resolution during those hours. Thus, we have rewritten this paragrath (and subsequent analysis) (See L421-430).

L420: I suggest merging this paragraph with the one above

**Response**: agreed (See L435-450).

L424: merge with previous paragraph… same topic..
Response: agreed (See L435-450).

Section 5.2 how are biogenic emissions treated in LOTOS-EUROS? The background is also CAMS?

Response: We agree that this is not explained properly here. Yes, LOTOS-EUROS also uses CAMS data for the boundary conditions of chemistry. Regarding biospheric fluxes, these are not simulated internally in LOTOS-EUROS but read in from external datasets. In the CoCO2 project setup, biogenic fluxes from the VPRM (Vegetation Photosynthesis and Respiration Model) dataset, provided by DLR (Deutsches Zentrum für Luft- und Raumfahrt, the German Aerospace Center), are used as offline input in LOTOS-EUROS. We have added this to the text (see L468-473).

Figure 6. add the names of the stations somewhere on the Figure.

Response: Agreed, the names of the stations have been added to the caption of this figure (see Figure 7).

L483: The CO2bg was not really computed by DALES right? Just taken from CAMS and transported in DALES?

Response: Yes, this is correct. CO2bg values are taken from CAMS, interpolated onto the LES grid, and transported by DALES as a passive tracer.

L484-485: Using anomalies instead of full simulated mole fractions. So, why do you include a background? Why not full mole fractions and leave the observations untouched? You could also just compare enhancements.. Not clear why this is selected.

Response: We disagree with the reviewer's suggestion because the background concentrations in our study are not constant. While the background (BG) has variability, this variability is somewhat suppressed due to the 6-hourly temporal resolution of the input CAMS data, but it is still considered. Since this work focuses on variability rather than the mean-level offset caused by the background or full mole fraction differences in the model, it makes sense to plot the deviation from the mean for BG, CO2sum, and LOTOS-EUROS CO2 time series.

It is important to note that for almost all the statistics presented in subsequent figures, extracting the mean level does not affect the assessment of variability. Only the Mean Bias Error (MBE), which is related to the mean level, is affected, as this measure evaluates the mean level. Therefore, for statistical analysis it makes sense to include the full mole fraction only due to MBE. For this reason, we decided to plot ΔCO2 for CO2bg, CO2sum, and LOTOS-EUROS CO2 at figures 8 and 10.

L486-487: first time NEE comes up. I assume this is the difference between soil resp and gpp in DALES. Not clear how is it for LOTOS-EUROS.

**Response:** Yes, NEE (Net Ecosystem Exchange) is the sum of soil respiration and NPP (Net Primary Production), with $CO_2$ photosynthesis being the negative component. In general, NEE is a flux, but here, since we analyze mole fractions, NEE represents the contribution of the biosphere to the local CO2 mole fraction, extracted from the CO2sum tracer. As mentioned earlier, unfortunately, we cannot validate or compare NEE between LOTOS-EUROS and DALES, as fluxes/NEE contributions are not available in the LOTOS-EUROS output (since LOTOS-EUROS does not have separate tracers for the different contributions to CO2 mole fraction as in DALES).

L488-489: Why you do a spatial operation to solve for temporal dynamics? Not clear. Maybe Im not getting what you are doing. I think what you want to say is: "DALES is sampled at the Cabauw tower interpolating horizontally and vertically to match the Cabauw measured CO2 profile." correct?

**Response**: Yes, this is correct. We have reconsidered this sentence to make it clearer. (See L580-584).

L490-495: I would expect some statistics like MBE, RMSE, R2, instead of this subjective description.

**Response**: Agreed. We have rewritten this paragraph based on updated Figure 8 and decreased the level of subjectiveness (See L584-605).

L501-503: see my comment above about the CAMS selection as background.

**Response**: As mentioned earlier, the bias analysis of CAMS EGG4 has been conducted by ECMWF, and biases are estimated to be up to 2.5%, with an average of around 1%. Additionally, due to the lower temporal resolution of CAMS, we anticipated some level of under or overestimation during the intermediate hours (bacause of linear interpolation).

L509-511: biases, you can quantify this, I suggest doing it. Overestimation by how much? you have numbers, use them.

**Response**: The biases have been analyzed and discussed in the ECMWF report. According to their findings, the biases in the CAMS EGG4 data can reach up to 2.5%, with an average bias of approximately 1%. We have provided these values based on their analysis to ensure consistency with the available analysis, precented in the ECMWF report ( see L536-538).

L513: what is this? vertical component? if this is not to blame why do you mention it? Confusing.

**Response**: The "vertical component" refers to the 3-D forest structure. However, we agree that it is not directly relevant here and we have omitted from the main text to avoid confusion.

L515-518: Sounds like methods. Also, you are bringing the statistics quite late in the Results.

**Response**: Agreed. We have added a methods subsection that describes the statistical methods used to evaluate the models against observations (See L474-487).

L519: Bootstrap of MAE? Not clear why this is done. Not mentioned in the Methods.

**Response**: We have added Bootstrap method description to the methods subsection (See L474-487).

L521-522: It is not surprising that CO2sum is better than CO2bg.. I would not even report this. It is expected.

**Response**: agreed. We have omitted this from the text.

L525-526: its quite surprising that the mesoscale model explains 12% more of the variability than DALES. Why then going to LES scales ? (I know you show this later, but during the first reading I was wondering about this because the good DALES performance is presented after this).

**Response**: We agree with the reviewer's suggestion to first present the comparison to urban Westmaas and Slufter, followed by the rural Cabauw tower measurements. Currently, in the main text, the comparisons to urban Westmaas and Slufter measurements come before the evaluation against rural observations from the Cabauw tower (see Sect. 8.2 and 8.3).

L520-535:  most of the statistics show LOTOS-EUROS is better than DALES.. why then using the LES? (Again this to show what the reader thinks when this is presented first). Additionally, I think you should fix the background and make the comparison using total mole fractions and not anomalies. Or explain clearly why the selection of the anomalies for this analysis.

**Response**: In this study, the main focus is to assess the variability of $CO_2$ mole fractions rather than the absolute values. By using deviation of the mean mole fractions, we aim to present the fluctuations in $CO_2$ concentrations, which is central to our analysis. Including the mean level is not necessary in this case, particularly in time series plots, where we are more interested in understanding the temporal variability, including contributions from anthropogenic emissions and Net Ecosystem Exchange (NEE). Thus, the protting of deviations is consistent with the primary objective of our study.

Figure 8. Why this plot shows full mole fractions? I suggest to be consistent either with anomalies like in Fig 7 or total mole fractions like this one.

**Response**: We appreciate the reviewer's comment. However, we respectfully disagree with the suggestion. As we mentioned early, in this study, the main focus is to assess the

variability of $CO_2$ mole fractions rather than the absolute values. By using deviation of the mean mole fractions, we aim to present the fluctuations in $CO_2$ concentrations, which is central to our analysis. Including the mean level is not necessary in this case, particularly in time series plots, where we are more interested in understanding the temporal variability, including contributions from anthropogenic emissions and Net Ecosystem Exchange (NEE). Thus, the protting of deviations is consistent with the primary objective of our study.

L539: you meant: "higher performance"? maybe change "higher" for "better"?

**Response**: Agreed. However, since we rewrote this paragraph, it is no longer included in the text.

L541-542: what if you compare gradients in both models? Maybe DALES is better than LOTOS here?

**Response** No, the gradient analysis shows similar $R^2$ and correlation at Cabauw.

L546-549: this sounds like methods, Maybe you can add a subSection in Methods called: "Model evaluation at atmospheric stations" and there tell the reader how do you sample DALES for each station and the height used.

**Response:** We have added a subsection outlining the methods used for statistical analysis (see L474-487).

Figure 9. Here it would be very interesting to have one single tracer for only anthropogenic emissions. You have one for NEE and not for the SNAP categories for which you have done most of the efforts processing data. With this you can easily argument that these stations are more affected by anthropogenic sources than Cabauw. The three tracers you have: *sum, *NEE and *bg don't allow to make such analysis.

**Response:** We have added the AE contribution time series in Figures 8 and 10.

L563: "Anthropogenic emissions dominate in the CO2 variability in this location", you could show this clearly if you would had an anthropogenic emission tracer. Now, is difficult to see if the model gets that.

**Response**: Agreed.

L551-576: A lot of subjective statements: slightly, typical, well-pronounce, etc. Only once numbers are used (L560).

**Response**: Agreed. We have rewritten the discussion for both Cabauw and Westmaas+Slufter and significantly reduced the level of subjectivity (see L530-555 and L584-605).

L578-579: Again, this (CO2sum better than CO2bg) is not surprising. I would not even report this.

**Response**: Agreed. We have excluded the comparison with CO2bg from this discussion (see L530-555 and L584-605).

L587-592: Not clear what the message is in this paragraph. First you refer to both models being off with low R2, then you refer to the normalized std being better in DALES. Then you go back to both models being off with a high RMSD. Think about a message and argument it. Now this paragraph is difficult to follow.

**Response**: agree, we rebuilt this paragrath to make it more easy to follow, first explaining dificulties (see 566-574).

Section 6.3: After reading this section, there was no clear message in the end. I had to go back and re-read to end up with no conclusive statement that summarizes the pros and cons at Wetsmaas and Slufter. You could refer to the difficulties the models have at Slufter, as you do now, but also make a conclusive statement for Westmaas. DALES was better there, so you can highlight this at the end of the Section.

**Response**: We agree with the reviewer's suggestion and have rewritten this paragraph, highlighting the strengths of DALES in reproducing $CO_2$ variability at Westmaas and discussing the challenges observed at Slufter (see L574-578).

L598-600: I had to wait 26 pages to see this. I think this should be presented earlier. Not sure if it is because the paper is too long, that it certainly feels quite late. I was wondering about the Anthropogenic emissions and their contribution from the beginning of the Results.

**Response**: We agree with the reviewer that the anthropogenic emission contribution should be introduced earlier. Therefore, we have added the AE contribution in Figures 8 and 10.

Figure 11. Nice plot. Some things to clarify:

- the % is taken from the full mole fraction or from dCO2?

**Response**: Yes, the percentage is taken from the full mole fraction of $CO_2$. We have clarified this in the caption of Figure 12.

- not clear if dCO2 is from the model or from the obs?

**Response**: $dCO_2$ is from the model. Observations are not shown here, as individual components cannot be derived from observations.

- if dCO2 is based on the model, then consider adding information on the dCO2 from the observations. You could do this by subtracting the background from the observations, making sure you have bias corrected CAMS if needed. See my comments above about this.

**Response**: Unfortunately, this is not possible, as the plot is designed to separate $CO_2$ contributions into its components, which cannot be done for the measurements. However, the comparison between observed and modeled $dCO_2$ is already presented in the plots comparing modeled vs. observed $CO_2$ mole fractions as deviations from the mean (see Figures 8 and 10).

- I dont understand why you first use anomalies, then for the scatter plots you use full mole fractions and now, deviations from background Why not anomalies here? Why dCO2 since the beginning? Please argument this.

**Response**: We subtract the background in this case to show the contribution of individual components to the total $CO_2$ mole fraction. In previous analyses, we focused on variability, which is why we initially plotted deviations from the mean for both observations and models. For statistical analyses, we used the full mole fraction since statistical metrics assessing variability are not affected by absolute values (only for MBE).

L628-629: This is true. But for making this statement you should show you are really doing this. I did not see a map (100x100m) of vegetation for your domain, it is not clear if you considered autotrophic respiration in DALES and there is no validation of NEE from DALES with eddy fluxes somewhere in NL. As you presented the Introduction and the Methods, the reader assumes the focus is on anthropogenic emissions and in the results there is a shift in focus to NEE.

**Response:** We agree that validating the biogenic fluxes from DALES would be a valuable addition, especially given their potential influence on $CO_2$ concentrations. While this is beyond the scope of the current study, we plan to address it in future work.

we have added a map of vegetation types to the paper (see Figure 1). However, validating biogenic fluxes is not feasible within our current setup, as our focus is on $CO_2$ concentrations rather than fluxes, which would require adjustments to the output procedure.

Regarding autotrophic respiration, it is accounted for in the Ags scheme. The Ags scheme has been previously evaluated (e.g., https://www.nature.com/articles/ngeo1554), though not specifically in an LES framework.

Section 7. What about the perspective of LES for atmospheric inversions? something to mention here? You can speculate on optimizing specific SNAP categories and use this as a benchmark for what the country or city is reporting (bottom-up).

**Response**: We agree with the reviewer that LES has strong potential for atmospheric inversions and can serve as a benchmark for optimizing specific SNAP categories and comparing bottom-up emission reports. We have added a paragraph discussing this perspective in the main text (see L724-727).

L645-649: I doubt higher resolution will be the main solution to the CAMS background. To get continental CO2 signals properly you need at least to optimize the fluxes (inversion) and have a good atmospheric transport. The CAMS version you are using doesn't do that, I believe so.. not clear either.

**Response**: We thank the reviewer for this comment. We agree that simply improving resolution may not be sufficient to properly capture continental $CO_2$ signals. To accurately represent these signals, it is essential to optimize fluxes using inversion techniques, along with a robust atmospheric transport model.

In our study, we used the CAMS EGG4 delayed mode, which incorporates bias corrections. This version addresses some of the limitations by adjusting for known biases in the model output. However, flux optimization is not performed within the CAMS EGG4 dataset. While other CAMS products do include flux optimization, their spatiotemporal resolution is lower. We agree that a more comprehensive approach, involving flux inversions and improvements in atmospheric transport, would significantly enhance the accuracy of $CO_2$ signals.

L689: access or link to code for downscaling, not seen so far.

**Respose**: yes, the link to downscaling code has been added to the emission section (see L300-301).

L704. "also identified limitations of the current framework" like?

**Response:** yes, we aggee that the phasing is not well formulated. We have ephased this sentence in the main text: Besides, we identified limitations in the current framework, such as larger deviations from observations at the Cabauw rural location and challenges in simulating coastal environments, which will require further improvement in future developments. (See L759-761)

L717: "with inversion techniques" This should be mentioned earlier in the Perspective section. How and what is the feasibility and also what type of information can it bring.

**Response**: Thank the reviewer for the suggestion. We agree and have now addressed the role of inversion techniques earlier in the Perspective section, discussing their feasibility and potential to improve $CO_2$ flux estimation. Specifically, we highlight how inversion methods can enhance emission quantification and support the independent validation of national inventories (see L724-727).

L718: "sub-scale processes" you meant sub-grid? also give examples.

**Response**: We agree, and we have rewritten the sentence as follows: "Furthermore, by delivering detailed information on sub-grid processes, such as turbulence, boundary layer dynamics, and localized emission dispersion, DALES can enhance parameterizations in larger-scale models through nesting, contributing to more accurate regional climate predictions." (see L774-777).

L720-721: "will support more informed decision-making" How? I don't see this for the near future with a couple of days of simulations. I expect to see discussion on what are the perspectives of making longer simulations for at least 6 months so that they are relevant to policy.

**Response**: We agree that the current phrasing may not fully capture the potential. By 'supporting more informed decision-making,' we refer to the possibility of using LES results to downscale larger-scale models (such as LOTOS-EUROS) in the future, potentially with the assistance of AI techniques (we have added this example at L777). This approach could improve the accuracy of long-term forecasts, which would be more relevant for policy development. Although the current simulations are short-term, extending them to several months would provide more valuable data for policy-related decisions. With advancements in computing power and storage, extending the simulation timeframe to longer periods could become feasible.

Code and data availability: the offline workflow repo should be referenced in the main text. The DALES 4.4 repo should also be reference earlier.

**Respose**: agreed, we have added these references to the main text (see L141 and L300-301).

Table A1. I suggest you add this on the main text, together with a description if they are considered point or area sources.

**Response**: Agreed. We have moved this table to the main text (see L118-119).

---

## Author Response (AR2)

Response to Referee #2 report

Dear Referee,

We thank to Referee #2 for re-assessing our manuscript and for additional suggestions and corrections to improve our manuscript. Our detailed responses to each comment and suggested correction can be found below (in blue):

The manuscript "Carbon dioxide plume dispersion simulated at hectometer scale using DALES" by Karagodin-Doyennel et al. presents a well-executed study using the LES model DALES at 100×100 m resolution to simulate CO2 mole fractions over a region in the Netherlands. The model includes both anthropogenic and biogenic fluxes, with a clear method for downscaling emissions and CAMS forecasts providing the CO2 background. Model results are evaluated at three Dutch sites—Cabauw, Slufter, and Westmaas—using tagged tracers to analyze variability drivers and the contribution of specific flux components to measurements. I think the paper can be accepted after revising the following:

1. The short discussion (L724-727) about the perspective of using this set-up in an atmospheric transport inversion should be expanded. At the moment the these lines are quite vague and miss the chance to provide more details on what are the main challenges for achieving a high resolution emission estimate based on a top-down approach using LES that can be a benchmark for bottom-up estimates. For example, what is now the bottleneck to achieve this? Would an analytical inversion be feasible? I think a little more substance to this part can be added just with a few sentences.

Response: We agree with reviewer that it makes sense to expand this paragraph to clarify the potential and limitations of using LES in atmospheric transport inversions. Here is the revised version (L722-733):

In addition, LES can optimize emissions for specific SNAP categories by integrating top-down atmospheric observations with bottom-up inventories. This approach refines the spatial and temporal distribution of emissions, providing a high-resolution benchmark for validating and adjusting reported estimates. However, the use of LES in atmospheric transport inversions poses several challenges. This includes the limited spatial and temporal extent of LES domains that constrains inversion to local or short-term events. This requires careful nesting within larger-scale models to capture background conditions accurately (see, e.g., Barlow et al., (2011), Lauvaux et al., (2016)). Besides, high computational costs also constrain the ensemble size and averaging periods required for robust inversions. Although analytical inversions may be feasible for a limited number of tracers or emission parameters, the high dimensionality and inherent nonlinear dynamics of LES simulations generally require ensemble-based methods (e.g., Brunner et al., (2019)). Despite these challenges, LES-based inversions offer a valuable framework for process-level understanding and can serve as a benchmark for evaluating bottom-up inventories under well-constrained conditions.

2. I suggest the authors to perform another proofreading to correct some grammar typos or wording of some parts of the text, for example:
- Abstract title is repeated

Respose: Agreed. Redundant "Abstract" word has been deleted.

- L182. latheral -> lateral
Respose: Agreed. "Latheral" has been replaced with "lateral".

- L363, merge this sentence with previous paragraph
Respose: Agreed. This sentence has been merged with previous paragrath.

- Figure 6. Horisontal -> Horizontal

Response: The Figure 6 has been corrected and updated (see L450-451).

- L640. Distinct compounds -> individual flux components

Response: Agreed. The "Distinct compounds"has been replaced with "individual flux components"

- L683. To achieve this -> Achieving this will require..

Response: Agreed. "To achieve this " has been replaced with "Achieving this"

- L699-L701. Merge with previous paragraph
Response: Agreed. This sentence has been merged with previous paragrath.

- L720. First sentence sounds repetitive, deals with the same topic as the paragraph above.
Response: Agreed. This sentence has been removed.

- L740. I suggest to remove this sentence. Already clear from the discussion above.
Response: Agreed. This sentence has been removed.

Additionally, as suggested, we reviewed the manuscript and made several revisions to improve language and phrasing.